# Brain-to-cervical lymph node signaling after stroke

Elga Esposito[1], Bum Ju Ahn[1], Jingfei Shi[1,2], Yoshihiko Nakamura [1,3], Ji Hyun Park[1], Emiri T. Mandeville [1], Zhanyang Yu[1], Su Jing Chan [1,4], Rakhi Desai[1], Ayumi Hayakawa[1], Xunming Ji[2], Eng H. Lo[1,5]* & Kazuhide Hayakawa[1,5]*

After stroke, peripheral immune cells are activated and these systemic responses may amplify brain damage, but how the injured brain sends out signals to trigger systemic inflammation remains unclear. Here we show that a brain-to-cervical lymph node (CLN) pathway is involved. In rats subjected to focal cerebral ischemia, lymphatic endothelial cells proliferate and macrophages are rapidly activated in CLNs within 24 h, in part via VEGF-C/VEGFR3 signalling. Microarray analyses of isolated lymphatic endothelium from CLNs of ischemic mice confirm the activation of transmembrane tyrosine kinase pathways. Blockade of VEGFR3 reduces lymphatic endothelial activation, decreases pro-inflammatory macrophages, and reduces brain infarction. In vitro, VEGF-C/VEGFR3 signalling in lymphatic endothelial cells enhances inflammatory responses in co-cultured macrophages. Lastly, surgical removal of CLNs in mice significantly reduces infarction after focal cerebral ischemia. These findings suggest that modulating the brain-to-CLN pathway may offer therapeutic opportunities to ameliorate systemic inflammation and brain injury after stroke.

---

[1] Neuroprotection Research Laboratory, Departments of Radiology and Neurology, Massachusetts General Hospital and Harvard Medical School, Charlestown, MA, USA. [2] China-America Institute of Neuroscience, Xuanwu Hospital, Capital Medical University, Beijing, China. [3] Department of Emergency and Critical Care Medicine, Fukuoka University Hospital, Jonan, Fukuoka, Japan. [4] Department of Pharmacology, Yong Loo Lin School of Medicine, National University Health System, National University of Singapore, Singapore, Singapore. [5] These authors jointly supervised this work: Eng H. Lo, Kazuhide Hayakawa. *email: Lo@helix.mgh.harvard.edu; khayakawa1@mgh.harvard.edu

nflammation after stroke is highly complex. Multifactorial signals can be produced by damaged brain cells and these may act as chemoattractants and activators of granulocytes, macrophages, and other inflammatory cells[1,2]. After CNS injury or disease, peripheral circulating immune cells are rapidly activated, cross the blood brain barrier, and influence injury and recovery[3–5]. For example, release of cytokines such as IL-6 and TNF-alpha from peripheral blood cells were increased in patients after stroke[6]. These inflammatory signals in the periphery are then thought to regulate the complex immune response in the CNS. After stroke, leukocytosis is associated with stroke severity, infarct volume, and worse functional outcomes[7–9], whereas, under some circumstances, monocyte-derived macrophages can mediate phagocytosis for clean-up and tissue remodeling during the recovery phase after stroke[10]. In general, infiltrated immune cells contribute to secondary injury but it is now recognized that both negative and positive effects may take place. However, what remains unknown is how the damaged brain sends the initial signal to trigger peripheral responses in the first place.

Cerebrospinal fluid (CSF) is produced by the choroid plexus and flows through ventricles to reach the subarachnoid space and dural venous sinuses[11,12]. Historically, arachnoid villi in the superior sagittal sinus are known as outflow sites for absorption of the CSF into the blood stream[13–15]. More recently, extensive studies in many different species have indicated a role for additional routes for CSF drainage, whereby CSF, interstitial fluid (ISF), cells, and soluble antigen drain from the subarachnoid space to the cribriform plate and nasal mucosa in animals and humans[16–19]. In rabbits and sheep, these lymphatic-like systems absorbed 30–50% of total outflow, while the rest might be drained through arachnoid villi[20,21].

Drainage from CNS into cervical lymph nodes (CLNs) has been documented in multiple studies utilizing tracers or antigen injection into CSF[22–24]. Lymphatic-like structures in meninges were described in adult mouse brains and this pathway may carry extracellular solutes into cervical nodes[25–27]. The CNS drainage into lymph nodes may be involved in tissue inflammation since it has been reported that excision of the CNS-draining lymph nodes reduced the pathology of experimental autoimmune encephalomyelitis within the spinal cord[28].

In this proof-of-concept study, we use rat and mouse models of focal cerebral ischemia along with co-cultures of lymphatic endothelium and macrophages to show that (a) VEGF-C/VEGFR3 signaling activates lymphatic endothelium in CLNs after ischemic stroke, and (b) pharmacological blockade of VEGFR3 signaling or surgical removal of superficial CLNs ameliorates post-stroke inflammation and reduces brain injury. These data suggest that brain-to-CLN signaling may be responsible for triggering systemic inflammatory responses after acute stroke. Further molecular dissection of this pathway may lead to novel therapeutic approaches for ischemic stroke.

## Results

**Lymphangiogenesis in CLNs after stroke**. In SD rats, injection of Evans blue into the lateral ventricles or brain parenchyma led to the rapid appearance of the dye in CLNs; dye-derived fluorescence was not detectable in axillary lymph nodes (ALNs) or inguinal lymph nodes (ILNs) (Fig. 1a, Supplementary Fig. 1a). Is it possible that after cerebral ischemia, this brain-to-CLN pathway may allow the damaged brain to rapidly send signals to the systemic immune system? To test our hypothesis, we induced focal cerebral ischemia in SD rats by transiently occluding their middle cerebral arteries (MCA) for 100 min, then removed deep and superficial CLNs and ILNs for analysis at 3 and 24 h.

Immunohistochemistry showed co-localization of lymphatic endothelial markers such as LYVE-1, Podoplanin, and VEGFR3 (Supplementary Fig. 1b). Double-staining using LYVE-1 and Ki67 antibodies confirmed that lymphatic endothelium in the area of subcapsular sinus in superficial CLNs were increased as early as 3 h after focal ischemia, and the lymphatic response in superficial CLNs was sustained until 24 h after focal ischemia (Fig. 1b–c). Proliferation of lymphatic endothelial cells was also observed in deep CLNs, but it was not statistically significant (Fig. 1b–c). Lymphatic endothelial proliferation did not appear to be induced in ILNs (Fig. 1d). VEGFR3 regulates lymphatic endothelial proliferation and lymphangiogenesis[29,30]. Therefore, we assessed VEGFR3 activation. Western blots following VEGFR3 immunoprecipitation showed that VEGFR3 was phosphorylated in superficial CLNs after cerebral ischemia, along with upregulation of the prototypical receptor ligand VEGF-C in CSF (Fig. 1e–f, Supplementary Fig. 1c). Intra-cerebroventricular injection of the VEGFR3 tyrosine kinase inhibitor MAZ51 decreased lymphatic endothelial proliferation in the superficial CLNs after focal cerebral ischemia (Fig. 1g–i).

**Activation of CLN macrophages after stroke**. LN macrophages are directly exposed to lymph fluid. Therefore, we also assessed macrophage response to cerebral ischemia. LN macrophages were detected by two different markers, CD169 and CD68. CD169 is a marker for macrophages in the subcapsular sinus and medullary sinus[31], and CD68 is expressed in macrophages present in the follicles, interfollicular zone, and sinus in LNs[32]. While lymphatic endothelial proliferation was rapidly increased in superficial CLNs after focal cerebral ischemia, CD169-positive macrophages in the subcapsular sinus were modestly proliferated (Fig. 2a–c). However, FACS analysis showed that focal cerebral ischemia appeared to increase IL-1β expression in CD68-positive macrophages, and treatment with MAZ51 reduced IL-1β-positive macrophage pools within superficial CLNs at 24 h after focal cerebral ischemia (Fig. 2d–e). Moreover, confocal microscopy analysis demonstrated that CD169-positive macrophages in the subcapsular sinus also strongly expressed IL-1β; MAZ51 treatment decreased these co-localization signals at 24 h after focal cerebral ischemia (Fig. 2f–g).

**Microarray analysis of CLNs after stroke**. To further investigate this phenomenon of CLN activation after cerebral ischemia, we examined a second species. Similar to rats, Evans blue dye tracing confirmed the presence of brain-to-CLN connection in C57BL6 mice (Supplementary Fig. 2a–b). After focal cerebral ischemia, CSF, deep and superficial CLNs, ALN, and the spleen were extracted for analysis. VEGF-C appeared to be elevated in CSF after cerebral ischemia (Supplementary Fig. 2c). In CLNs, expression levels of the lymphatic endothelium marker LYVE-1 were increased after cerebral ischemia, superficial CLNs showed slightly more robust responses compared to deep CLNs (Fig. 3a). In contrast, no clear changes were detected in ALNs or the spleen (Fig. 3a). Concomitantly, immunohistochemistry revealed an increase of lymphatic vessel intensity (Fig. 3b–c, Supplementary Fig. 2d). Micro-array analysis of LYVE-1-positive cells isolated from CLNs at 3 h after reperfusion showed that their transcriptome was rapidly altered (Fig. 3d–e). Gene Set Enrichment Analysis (GSEA) suggested that differentially expressed genes were largely related to matrix pathways and transmembrane receptor protein tyrosine kinase activity (Fig. 3f, Supplementary Fig. 3). The upregulated genes included *Fkbp5, Hdc, Sesn1, Lyve1, Cyp2b10, Dccp2, Flt1, Ccl28, IL1b, Kdr*, and *Flt4* (depicted in annotated volcano plot, Fig. 3g).

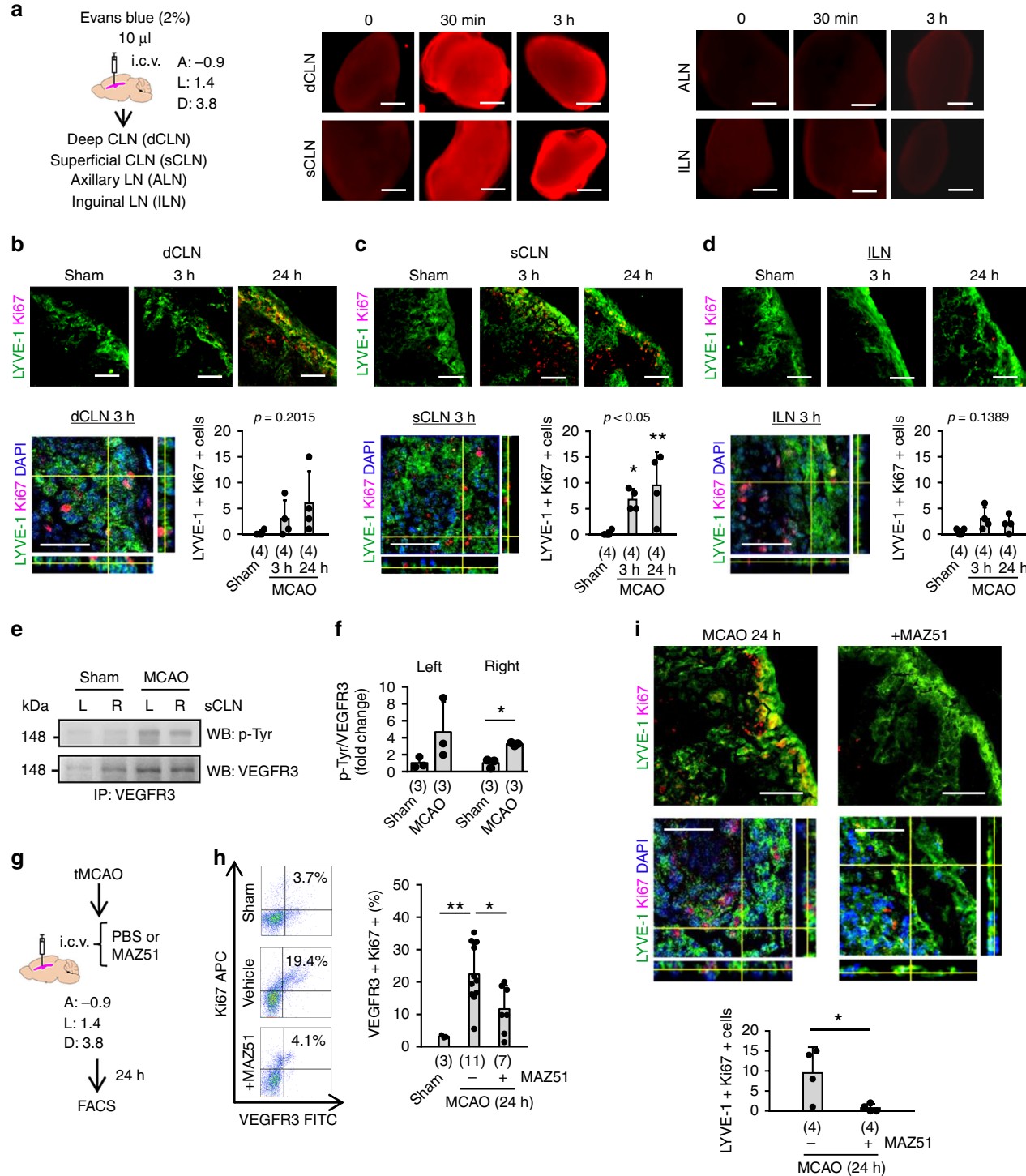

**Blocking VEGFR3 reduces CLN inflammation**. Next, we asked whether, similar to our rat model findings, a blockade of VEGFR3 tyrosine kinase may also interfere with the hypothesized brain-to-CLN pathway in mice. Once again, focal cerebral ischemia was induced in C57BL6 mice. Then the mice were randomly treated, immediately after reperfusion with either MAZ51 or PBS vehicle administered via intranasal cavity (Fig. 4a, Supplementary Fig. 4). Although systemic drug effects cannot be unequivocally excluded, many other pharmacology studies have used this intranasal route to target cervical nodes[33–35]. Flow cytometry analysis demonstrated that MAZ51 treatment significantly suppressed tyrosine phosphorylation in post-stroke CLN lymphatic endothelium, and

decreased pro-inflammatory macrophages (Fig. 4b–c, Supplementary Fig. 5). Concomitantly, immunohistochemistry demonstrated that macrophages in the subcapsular sinus in deep and superficial CLNs increased TNF-α expression after focal ischemia, this response was suppressed by MAZ treatment. Instead, ILNs did not show significant responses to cerebral ischemia and MAZ51 treatment (Fig. 4d–f). We confirmed TNF-α expression in CD169-positive macrophages in superficial CLNs by confocal microscopy (Fig. 4g–h). Next, we quantified cytokine/chemokine expressions in LNs by western blot analysis. After focal ischemia, pro-inflammatory cytokines, including TNF-α and IL-1β, were increased in superficial CLNs but not in ILNs, and the blockade of

**Fig. 1** Cervical lymph node activation in rats after stroke: **a** Evans Blue dye (2%, 10 µL) was injected into lateral ventricles in normal male Sprague Dawley (SD) rats. Within 30 min, Evans Blue fluorescence was detected in deep and superficial cervical lymph nodes (CLNs) but not in axillary lymph nodes (ALNs) or inguinal lymph nodes (ILNs). Scale: 500 nm. **b–d** Male SD rats were subjected to 100 min of transient focal cerebral ischemia. Deep and superficial CLNs were collected and analyzed by immunohistochemistry. We did not count LYVE-1 or Ki67 single-positive cells to exclude potential signal from the macrophage population. **b** Proliferation of lymphatic endothelial cells were also observed in deep CLNs, but it was not statistically significant. **c** LYVE-1 positive lymphatic endothelial cells in the area of subcapsular sinus in superficial CLNs rapidly proliferated as early as 3 h until 24 h after focal ischemia ($n = 4$ biologically independent animals). **d** At the same time, ILNs were collected as a negative control. Note: Some of Ki67 could be co-expressed with immune cells including macrophages and T cells[58]. *$P < 0.05$, **$P < 0.01$ vs Sham, one-way ANOVA followed by Fisher's LSD test. Scale: 100 nm. **e, f** Western blot, following immunoprecipitation, demonstrated that VEGFR3 was phosphorylated in superficial CLNs at 24 h after transient focal ischemia ($n = 3$). *$P < 0.05$, unpaired $t$-test. **g** PBS (5 µL) or MAZ51 (50 ng/5 µL) was injected into lateral ventricles immediately after reperfusion and FACS analysis was performed at 24 h. **h** FACS analysis indicated that focal ischemia promoted the proliferation of VEGFR3 positive cells in the superficial CLNs. Intracerebroventricular injection of MAZ51 significantly suppressed the proliferation of VEGFR3-positive cells in CLNs (Sham; $n = 3$, MCAO; $n = 11$, MCAO + MAZ51; $n = 7$ biologically independent animals). *$P < 0.05$, **$P < 0.01$, one-way ANOVA followed by Fisher's LSD test. **i** Immunohistochemistry demonstrated that MAZ51 treatment decreased the proliferation of LYVE-1 positive lymphatic endothelium that was observed after ischemia ($n = 4$). Scale: 100 nm. **$P < 0.01$, unpaired $t$-test. All values are mean +/− SD.

VEGFR3 tyrosine kinase with MAZ51 treatment significantly decreased TNF-α and IL-1β upregulation in superficial CLNs (Fig. 4i–l). In addition, analysis of the full micro-array dataset revealed that CCL28 was the most significantly altered chemokine in the CLN after cerebral ischemia. This observation may be consistent with the fact that CCL28 is known to regulate lymphatic endothelial migration[36]. CCL28 was increased in superficial CLNs but not in ILNs, and the blockade of VEGFR3 tyrosine kinase with MAZ51 treatment significantly reduced CCL28 in superficial CLNs after cerebral ischemia (Fig. 4i–l). We also analyzed the prototypical chemokine CCL2 since it has been well studied in experimental and clinical stroke[37,38]. Western blot analysis confirmed that CCL2 was also significantly increased in post-stroke superficial CLN, accordingly, and MAZ51 treatment reduced CCL2 expression (Supplementary Fig. 6).

**Lymphatic endothelium increases pro-inflammatory macrophages**. To further investigate this signaling mechanism, we turned to cell culture systems. Lymphatic endothelial cells were isolated from mouse cervical nodes by LYVE-1-conjugated magnetic beads (Fig. 5a). Immunostaining confirmed that these cells were positive for VEGFR3, LYVE-1, Podoplanin and CD31, but negative for CD45 and CD11b (Supplementary Fig. 7a–c). Quantitative PCR analysis further verified that another lymphatic endothelial marker, *Prox1*, was highly expressed in the isolated cells, but not in peritoneal macrophages (Supplementary Fig. 7d). Next, we co-cultured lymphatic endothelium with primary mouse macrophages in a standard transwell system, and then we stimulated VEGFR3 in the lymphatic endothelial cells by adding VEGF-C to the upper chamber (Fig. 5b). This in vitro system demonstrated the VEGFR3-activated lymphatic endothelium was able to induce an upregulation of pro-inflammatory signals such as iNOS and IL-1β in macrophages within the lower chamber (Fig. 5c–f). As a control, adding VEGF-C directly to macrophages did not induce any detectable iNOS and IL-1β activation (Fig. 5g–i). Altogether, these cell culture experiments may be consistent with the idea that VEGFR3 signaling activates lymphatic endothelium and thereafter, these activated lymphatic cells may then upregulate potentially pro-inflammatory macrophage responses.

**Blockade of VEGFR3 reduces brain injury after stroke**. Our results showed, so far, that after focal cerebral ischemia, CLNs are activated and that blocking VEGFR3 tyrosine kinase reduces CLN inflammation. To further investigate inflammation in the brain, C57BL6 mice were subjected to 60 min of transient focal cerebral ischemia and then treated with either PBS vehicle or MAZ51. We performed collagen IV (basal lamina) or vWF (endothelium)

immunostaining to provide landmarks for visual context of the immune cell accumulation in the brain after focal cerebral ischemia. As expected, immunohistochemistry confirmed that monocytes/macrophages and neutrophils were accumulated in ischemic areas and the peri-infarct cortex along with some T-cell infiltration at 72 h after stroke (Fig. 6a–b). Immune cells were also found in close proximity to leptomeninges, choroid plexus, and brain parenchyma in the ipsilateral hemisphere (Supplementary Fig. 8), consistent with emerging literature supporting a role for meningeal vessels as a route for immune cell infiltration and post-stroke inflammation[39–41]. MAZ51 treatment significantly decreased the accumulation of pro-inflammatory F4/80+/CD16+ monocytes/macrophages without any detectable changes in IgG leakage (Fig. 6c–e). But most importantly, MAZ51 significantly reduced the volume of brain infarction (Fig. 6f). Taken together, these data suggest that lymphatic VEGFR3 signaling may amplify brain inflammation and damage after cerebral ischemia.

**Lymphadenectomy reduces brain injury after stroke**. Finally, we asked whether surgical interruption of this pathway could substantially alter stroke outcomes. Superficial CLNs were dissected and removed right before transient focal cerebral ischemia in C57BL6 mice. Then immune cell response in systemic circulation, the brain, and infarct volumes were examined (Fig. 7a). Flow cytometry analysis demonstrated that cervical node lymphadenectomy ameliorated the response of neutrophils and macrophages in both blood and the brain (Fig. 7b–c, Supplementary Fig. 9). Along with these alterations in immune cell response, removal of superficial CLNs also significantly decreased pro-inflammatory macrophages and cerebral infarction at 72 h after stroke, although no clear differences were detectable in IgG leakage (Fig. 7d–f). As a control, surgical removal of ILN did not reduce infarct volume (Fig. 7g).

**Discussion**

Taken together, these in vitro and in vivo findings in both rat and mouse models support the overall idea that brain-to-CLN signaling via VEGFR3 tyrosine kinase may be involved in the connection between central brain injury and peripheral immune activation. This crosstalk mechanism may acutely worsen brain damage after cerebral ischemia (Fig. 7h).

It has been proposed that beyond cell-cell signaling within the brain per se, dynamic crosstalk between brain and systemic responses, such as circulating blood cells, may also be important[42]. After CNS injury or disease, systemic immune responses may influence injury and recovery. What may be missing from the collective literature to date is how the stroke-damaged brains initially send out signals to trigger systemic inflammation. Our

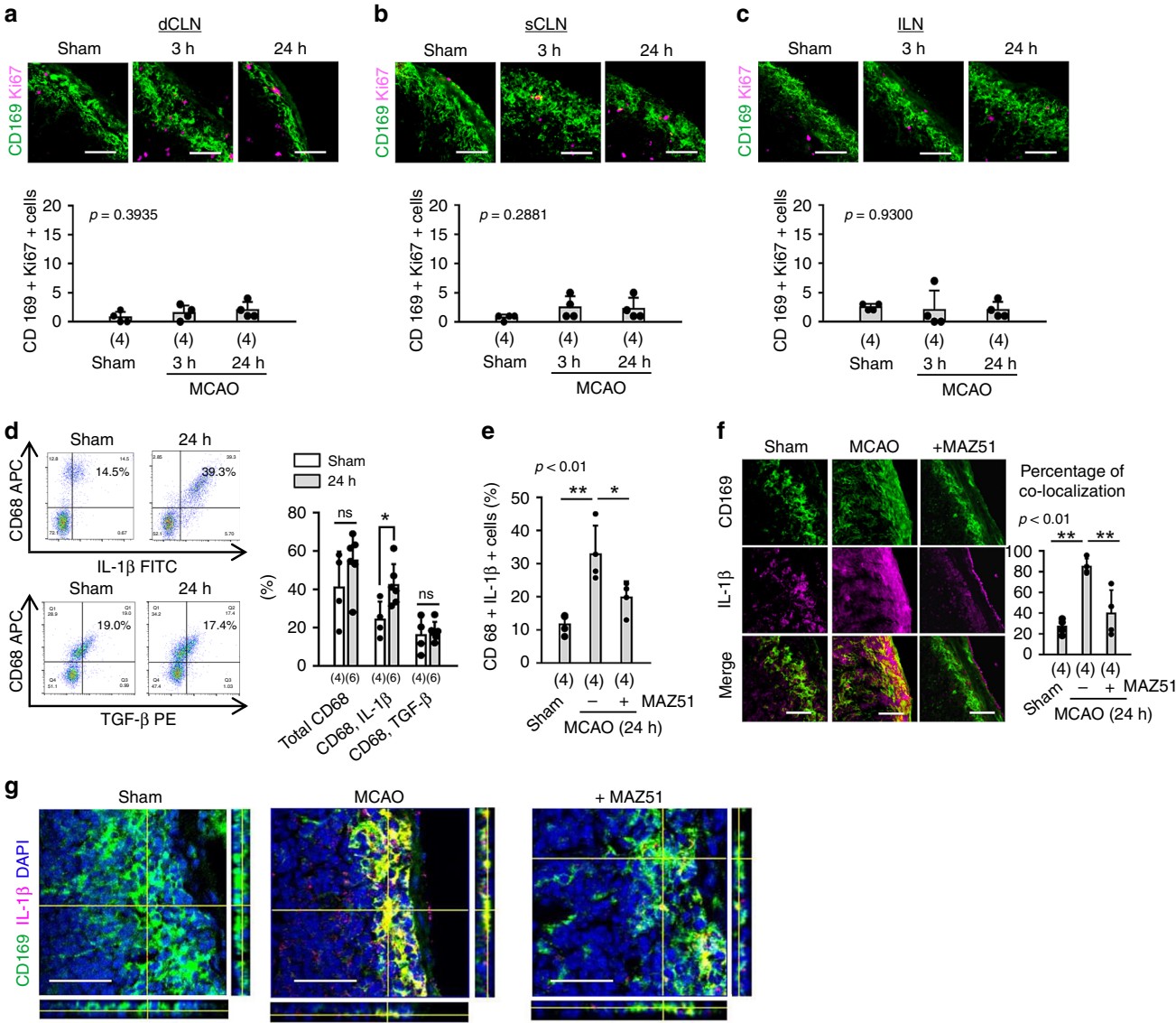

**Fig. 2** Macrophage activation in rat CLNs after stroke: **a–c** Male SD rats were subjected to 100 min of transient focal cerebral ischemia. Deep and superficial CLNs, and ILNs were collected and analyzed by immunohistochemistry. Lymph node resident CD169 positive macrophages co-labeling with Ki67 were identified as proliferating cells. Immunostaining showed that macrophage proliferation was slightly but not significantly increased in CLNs after focal cerebral ischemia ($n = 4$ biologically independent animals). Scale: 100 nm. One-way ANOVA followed by Fisher's LSD test. **d** Inflammatory factors, including prototypical cytokines such as interleukin-1β (IL-1β) and transforming growth factor beta (TGF-β), were analyzed, by flow cytometry analysis, in CD68 positive macrophages. After stroke, IL-1β positive macrophages were increased in superficial CLNs, but TGF-β positive subsets did not change (Sham; $n = 4$, MCAO; $n = 6$ biologically independent animals). *$P < 0.05$ vs Sham, one-way ANOVA followed by Fisher's LSD test. **e** Intracerebroventricular injection of MAZ51 (50 ng/5 μL) significantly reduced IL-1β positive macrophages ($n = 4$ biologically independent animals). *$P < 0.05$, **$P < 0.01$, one-way ANOVA followed by Fisher's LSD test. **f, g** Immunohistochemistry confirmed that CD169 positive macrophages expressed IL-1β after focal cerebral ischemia and MAZ51 treatment reduced IL-1β positive macrophages in subcapsular sinus. Scale: 100 nm. All values are mean +/− SD.

present study provides proof-of-principle that brain-to-CLN signaling may be the underlying pathway involved. We found that after focal ischemia, the damaged brain rapidly secreted VEGF-C into CSF and activated lymphatic endothelium in superficial CLNs through VEGFR3 phosphorylation. Importantly, blockade of VEGFR3 phosphorylation or superficial CLN lymphadenectomy ameliorated acute brain infarction without causing deleterious effects at a later point in time. There may be ongoing discussions as to the extent and depth of penetration of immune cells into the various levels of the neurovascular unit[43,44], but our study is not aimed to resolve these complex issues. The causal importance of cervical node signaling in stroke is supported by our primary finding, i.e. removal of superficial CLNs but not

other nodes, significantly reduced infarction after focal cerebral ischemia.

Nevertheless, a few caveats and questions should be kept in mind. First, our study is primarily based on a brain-to-CLN pathway that activates CLNs during the acute stage of stroke. Further studies are needed to verify the upstream details of lymphatic drainage that may likely occur via the cribriform plate or mucosal lymphatics[24,45,46], and ask whether other lymph nodes such as parotid nodes may also be involved. Second, our mouse model of focal ischemia showed rapid LYVE-1 upregulation in CLNs that may represent lymphangiogenesis and lymphatic endothelial proliferation[47]. However, some reports suggest that LYVE-1 is downregulated during inflammation in lymphatic

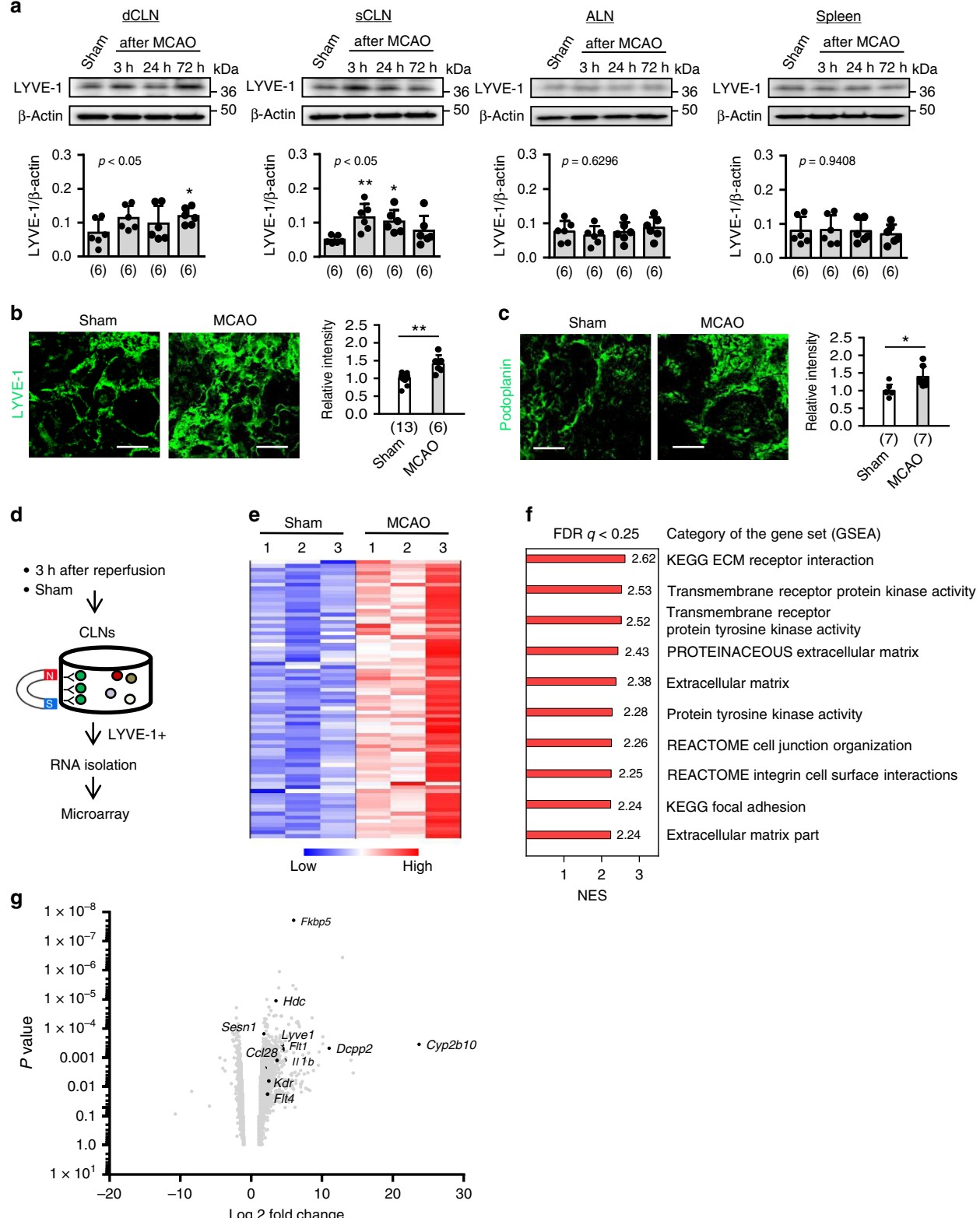

endothelium[48,49]. The process of lymphatic activation and lymphangiogenesis may be highly complex and context-dependent. Future studies should examine how CLN response is influenced by the type, severity, and timing of brain injury or disease, including detailed mapping of multiple lymphatic vessel markers including Prox1. Third, we removed only superficial CLNs because lymphatic endothelial proliferation in superficial CLNs was robust

compared to deep CLNs in rats and mice. Additionally, recent studies have suggested that drainage of brain signals (e.g. lactate) occurs in superficial CLNs[50]. However, we acknowledge that deep CLNs may also be important in the brain-to-CLNs signaling. The role of deep CLNs in stroke pathophysiology should be addressed in future studies. Fourth, although blocking VEGFR3 signaling or surgically removing superficial CLNs significantly decreased

**Fig. 3** Cervical lymphatic activation in mice after stroke: **a** Male C57BL6 mice were subjected to transient 60 min of focal ischemia. Note: the artery occlusion time was titrated to achieve equivalent levels of injury in mice and rats. Sixty min of transient middle cerebral artery occlusion in mice result in the same infarct volume as 100 min of occlusion in rats, i.e. approximately 30% of the ipsilateral hemisphere. Western blot demonstrated that LYVE-1 expression was significantly upregulated in CLNs. LYVE-1 upregulation was robust in superficial CLNs compared to deep CLNs after focal ischemia. There were no significant changes in axillary lymph nodes or spleen after focal ischemia ($n = 6$ biologically independent animals). *$P < 0.05$ vs Sham resulted from Fisher's LSD test. **b, c** Immunohistochemistry using LYVE-1 antibody (Sham; $n = 13$, MCAO; $n = 6$ biologically independent animals) and Podoplanin antibody (Sham; $n = 7$, MCAO; $n = 7$ biologically independent animals) confirmed that lymphatic endothelial intensity was increased in superficial CLNs at 24 h after focal cerebral ischemia. Scale: 100 nm. *$P < 0.05$, **$P < 0.01$, unpaired t-test. Images were taken in the area of subcapsular sinus. **d** Experimental design for microarray using lymphatic endothelium isolated from CLNs after focal ischemia. Male C57BL6 mice were subjected to transient focal ischemia. At 3 h after reperfusion, LYVE-1 positive lymphatic vessels were collected from CLNs pooled from 4 mice, and RNAs were extracted for microarray analysis. **e** Focal ischemia upregulated genes in lymphatic endothelium (red in heat maps, $n = 3$, total animal number = 12). **f** Gene Set Enrichment Analysis (GSEA) suggested that differentially expressed genes were largely related to matrix and transmembrane receptor protein tyrosine kinase pathways. **g** Volcano plot showed some upregulated genes after cerebral ischemia including *Fkbp5*, *Hdc*, *Sesn1*, *Lyve1*, *Cyp2b10*, *Dccp2*, *Flt1*, *Ccl28*, *IL1b*, *Kdr*, and *Flt4*. All values are mean +/− SD.

infarction after stroke, no statistically significant differences in neurological recovery were detected (Supplementary Fig. 10). Lymphatic inflammation can have complex biphasic actions; it can worsen acute tissue damage, but under some conditions it can also help resolve brain edema and promote delayed repair[51,52]. In our experiments, superficial CLN removal did not worsen neurological outcomes, suggesting that it does not interfere with endogenous recovery that is often present in rodent stroke models. Ultimately, lymphatic activation after CNS injury may regulate the balance between acute injury and delayed recovery, so this phenomenon should be carefully dissected. Fifth, we focused on macrophages. But of course, lymph nodes also contain T and B cells that are known to play key roles in stroke pathophysiology[53]. Further studies are warranted to investigate the entire spectrum of immune cells that are regulated by cervical lymphatics after stroke. Sixth, our results implicate a role for VEGFR3 signaling in CLN, and our initial data suggest that an increase of VEGF-C in the draining CSF may be the trigger. However, the damaged brain after stroke may produce many factors that can induce lymphatic inflammation, including NRP2[54], angiopoietin-1 and Tie-2[55], TGF members like BMP9-ALK1[56], and other DAMPs[57]. In a mouse model of experimental autoimmune encephalomyelitis (EAE), VEGF-C/VEGFR3 signaling increased lymphangiogenesis and CNS antigens in the draining lymph nodes. Inhibition of VEGFR3 with MAZ51 reduced CNS-derived antigen drainage in accompanied with decreasing EAE severity[58], suggesting that CNS antigens may also be involved in inflammatory responses in the lymph nodes. How lymphatic VEGFR3 mechanisms interact with other extracellular signals should be explored. Finally, although we document this brain-to-CLN mechanism in mice and rats, it remains to be fully confirmed whether it occurs in humans. How lymphatic drainage is able to refresh CSF in animals and humans is still a controversial topic. In mice, CSF appears to drain exclusively via lymphatics[59]. In humans, the question has never been answered directly. Nevertheless, there may already be indirect evidence to suggest clinical relevance. In stroke patients, neuronal glutamate receptor antigens and myelin basic protein fragments have been detected in CLNs[60], suggesting communication between brain and CLNs. Furthermore, emerging data from experimental models and clinical trials may now support the feasibility of directly injecting therapeutics into lymph nodes to block inflammation[61,62].

After stroke, the peripheral immune system is activated. Here, we provide evidence that a brain-CLN signaling pathway may be responsible. More broadly, lymphatic vascular inflammation represents an entirely separate field with its own biology and therapeutic advances[63]. Rigorously dissecting the pathophysiology of brain-lymphatic crosstalk may eventually lead us to novel mechanisms and targets for stroke.

## Methods

**Reagents**. Recombinant VEGF-C was purchased from Thermo Fisher Scientific (10–542-H08H5). Normal mouse IgG (sc-2025) and normal rabbit IgG (sc-2027) were purchased from Santa Cruz Biotechnology. VEGFR3 tyrosine kinase inhibitor, MAZ51 was purchased from EMD Millipore (676492–10MG).

**Antibodies**. Anti-β-actin (1:1,000, A5441, Sigma-aldrich), anti-p-Tyr antibody (1:500, sc-7020, Santa Cruz), anti-VEGF-C antibody (1:500, sc-374628, Santa Cruz), anti-VEGFR3 antibody (1:500, sc-365748, Santa Cruz), anti-iNOS antibody (1:500, ab3523, Abcam), anti-IL-1β antibody (1:500, ab9722, Abcam), anti-TNF-α antibody (1:200, GTX110520, GeneTex), anti-TGF-β antibody (1:200, ab64715, Abcam), anti-CCL28 antibody (1:500, MAB717, R&D systems), anti-Podoplanin antibody (1:200, sc-166906, Santa Cruz), anti-CD31 antibody (1:200, 550274, BD biosciences), anti-LYVE-1 antibody (1:500, NB100–725B, NOVUS biologicals), anti-vWF antibody (1:100, A0082, Agilent), FITC anti-CD4 antibody (1:100, 561828, BD Biosciences), FITC Neutrophil antibody (1:100, ab53453, Abcam), PE anti-CD34 antibody (1:100, 551387, BD Biosciences), APC anti-F4/80 antibody (1:200, 123116, BioLegend), FITC anti-CD16/32 antibody (1:200, 101306, BioLegend), PE anti-CD11b antibody (1:200, 557397, BD biosciences), anti-CD45 antibody (1:400, 20103–1-AP, Proteintech).

**Animals**. Male Sprague-Dawley rats (320–340 g) and male C57BL6 mice (24–27 g) were housed in pathogen-free facilities with 12 h day and night cycles. Experiments were performed under institutionally approved protocol in accordance with the National Institute of Health's Guide for the Care and Use of Laboratory Animals. All animals were randomly allocated to treatment groups.

**Focal cerebral ischemia**. All animals were anesthetized with isoflurane (1.5%) in 30%/70% oxygen/nitrous oxide. Transient focal ischemia was induced introducing a 6–0 (in mice) or 5–0 (in rats) surgical monofilament nylon suture (Doccol) from the external carotid artery into the internal carotid artery and advancing it to the branching point of the MCA. Cerebral blood flow was monitored, by continuous laser doppler flowmetry (LDF) (Perimed, North Royalton, OH, U.S.A.), in the area of the right MCA to confirm adequate occlusion. Animals that did not have a significant reduction to less than 30% baseline LDF values during MCAO were excluded. After occluding the MCA for 60 min in mice and 100 min in rats, the monofilament suture was gently withdrawn in order to restore blood flow, and LDF values were recorded for 10 min after reperfusion. Rectal temperature was monitored and maintained at 37 °C ± 0.5 °C with a thermostatically controlled heating pad during surgery and a heating lamp for 4 h after surgery. Functional outcome after stroke was assessed by a 10-point neurological severity score (NSS)[64]. The score consists of 10 individual clinical parameters, including tasks on motor function, alertness, and physiological behavior, whereby 1 point is given for failure of one task and no points are given for success. A maximal NSS of 10 points indicates severe neurological dysfunction, with failure at all tasks. In our hands, a mouse 60 min MCAO produces approximately 100 mm$^3$ or 30% infarction in a hemisphere as previously reported by other research groups[65,66]. Additionally, our previous studies demonstrate that brain damage severity after 60 min MCAO in a mouse is equivalent to one in 100 min MCAO model in rats[67–69].

**Evaluation of infarction and IgG leakage**. The animals underwent transcardial perfusion with phosphate-buffered saline (PBS). Brains were extracted, placed within a brain matrix and cut in 8 coronal sections of 1 mm each. The sections were then incubated in 2% TTC in 1× phosphate-buffered saline (PBS) for 10 min at 37 °C. Infarction volumes were quantified using the indirect morphometric method in order to reduce the effect of edema on infarct volume which is commonly used to evaluate infarct volume in the first 3 days after focal ischemia. Infarct volume (mm$^3$) was calculated by Image J software as follows; [the area

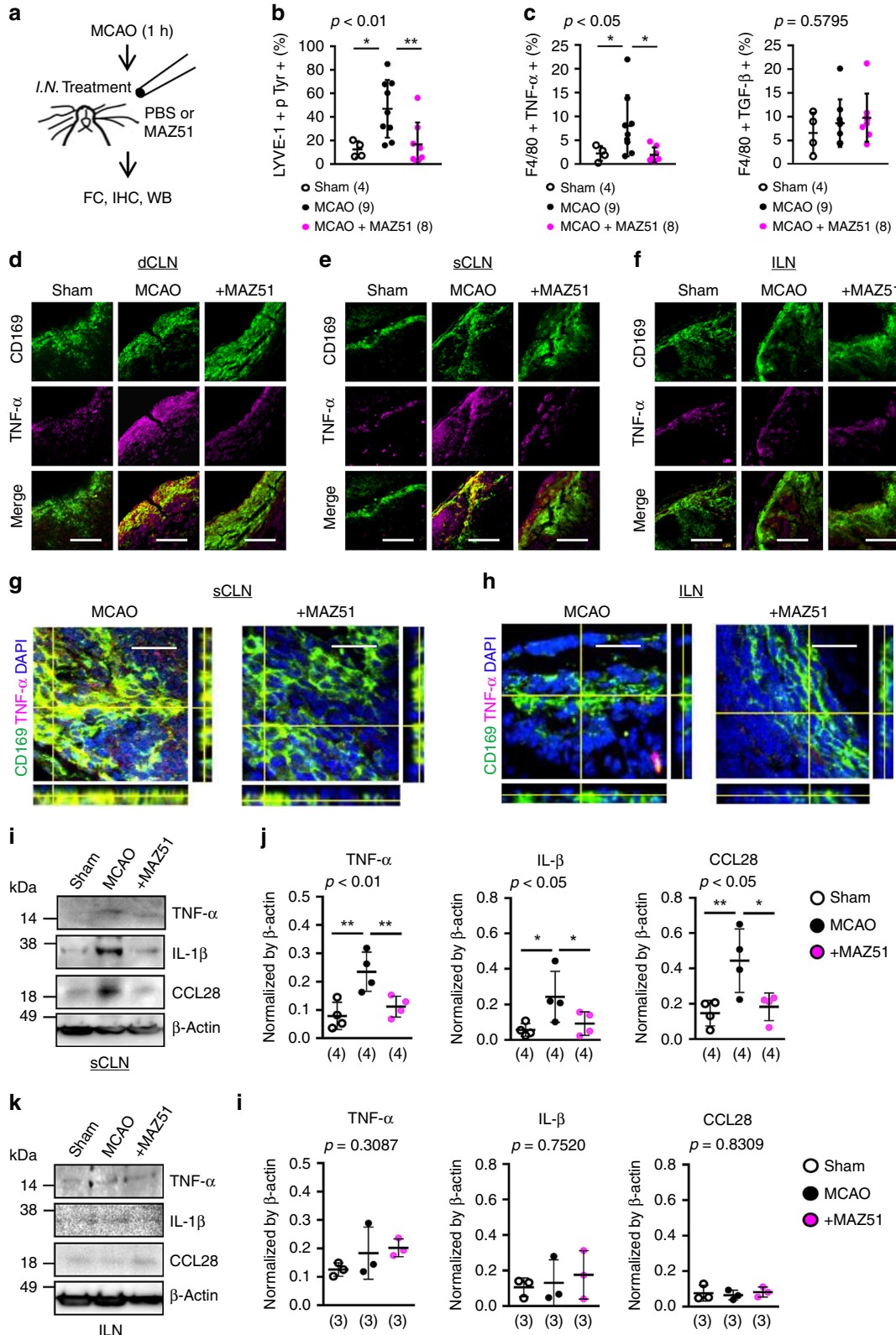

(mm²) of the contralateral hemisphere−the area (mm²) of the non-ischemic (healthy) tissue of the ipsilateral hemisphere] × 1 mm of each brain slice. Nissl staining was performed to define infarction at day 8 after focal ischemia. Fresh frozen brain sections were initially fixed with 4% paraformaldehyde for 10 min at room temperature. Then sections were processed with xylene (5 min), 95% ethanol (3 min), 70% ethanol (3 min), deionized distilled water (3 min), 1% w/v Crecyl Violet (8 min), distilled water (3 min), 70% ethanol (3 min), 95% ethanol (3 min), and 100% ethanol up to 10 dips, xylene (5 min). Xylene-based mounting solution

was dropped and coverslips were put on sections for imaging. IgG staining was performed by a combination of M.O.M kit (BMK-2202, Vector Laboratories) and ABC peroxidase staining kit (32020, Thermo Fisher Scientific). Fresh frozen brain sections were initially fixed with 4% paraformaldehyde for 10 min at room temperature. After blocking endogenous enzyme activity with BLOXALL blocking solution for 10 min, M.O.M mouse Ig blocking reagent was applied on sections for 1 h at room temperature, followed by washing sections with PBS twice. Then, biotinylated anti-mouse IgG was applied for 10 min. After washing sections with

**Fig. 4** Cervical lymphatic inflammation through VEGFR3 tyrosine kinase in mice after stroke: **a** Male C57BL6 mice were subjected to transient focal ischemia and, immediately after reperfusion the vehicle (PBS 10 μL) or MAZ51 (3 ng/10 μL) were injected into the nasal cavity. **b** MAZ51 treatment suppressed tyrosine phosphorylation in superficial CLN lymphatic endothelium at 72 h after focal ischemia (Sham; $n = 4$, MCAO; $n = 9$, MCAO + MAZ51; $n = 8$ biologically independent animals). *$P < 0.05$, one-way ANOVA followed by Fisher's LSD test. **c** FACS analysis demonstrated that MAZ51 treatment significantly decreased pro-inflammatory TNF-α positive macrophages in superficial CLNs; no clear changes were noticed in TGF-β positive macrophages. *$P < 0.05$, one-way ANOVA followed by Fisher's LSD test. **d–f**. Immunostaining revealed that CD169 positive macrophages increased TNF-α, while MAZ51 treatment decreased pro-inflammatory macrophages. Scale: 100 nm. **g–h** Confocal microscopy analysis demonstrated that TNF-α was highly co-localized with CD169 positive macrophages in the subcapsular sinus of superficial CLNs. MAZ51 treatment decreased the co-localization. TNF-α expression was not observed in ILN macrophages. Scale: 100 nm. **i–l** Western blot confirmed that TNF-α, IL-1β and CCL28 were significantly increased in post-stroke superficial CLNs. MAZ51 treatment reduced cytokine/chemokine expression ($n = 4$ biologically independent animals). ILNs weakly responded to focal cerebral ischemia as to TNF-α, IL-1β or CCL28 expression ($n = 3$ biologically independent animals). *$P < 0.05$, **$P < 0.01$, one-way ANOVA followed by Fisher's LSD test. All values are mean $+/-$ SD.

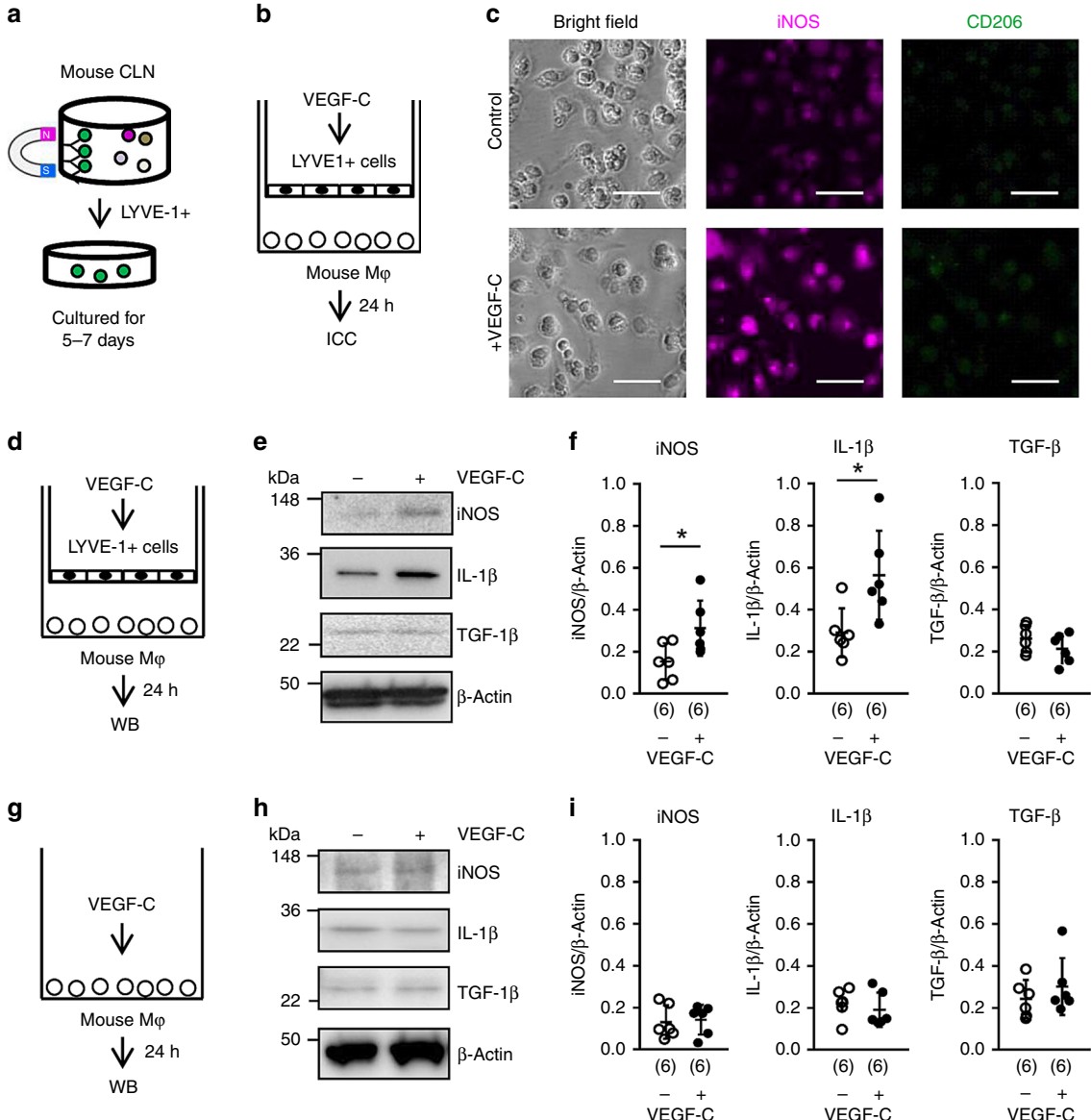

**Fig. 5** Activated lymphatic endothelium upregulates pro-inflammatory macrophages in vitro: **a** LYVE-1 antibody-conjugated magnetic beads were used for lymphatic endothelial isolation using CLNs isolated from 4 mice. **b** VEGFR3 was activated by adding VEGF-C (10 ng/ml) to mouse CLN lymphatic endothelial cells for 24 h, then cells were co-cultured with mouse peritoneal macrophages for another 24 h and analyzed by immunostaining. **c** Immunostaining showed that the expression of iNOS (M1-like) was upregulated, but CD206 (M2-like) expression did not change. Scale: 50 nm. **d–f** Western blot confirmed that iNOS and IL-1β were upregulated, but TGF-β did not change. *$P < 0.05$, unpaired t-test. **g–i** VEGF-C (10 ng/mL) was directly added to macrophages for 24 h. Western blot confirmed that VEGF-C itself did not change macrophage iNOS, IL-1β, and TGF-β expressions ($n = 6$ biologically independent cells, $n = 3$ biologically independent experiments), unpaired t-test. All values are mean $+/-$ SD.

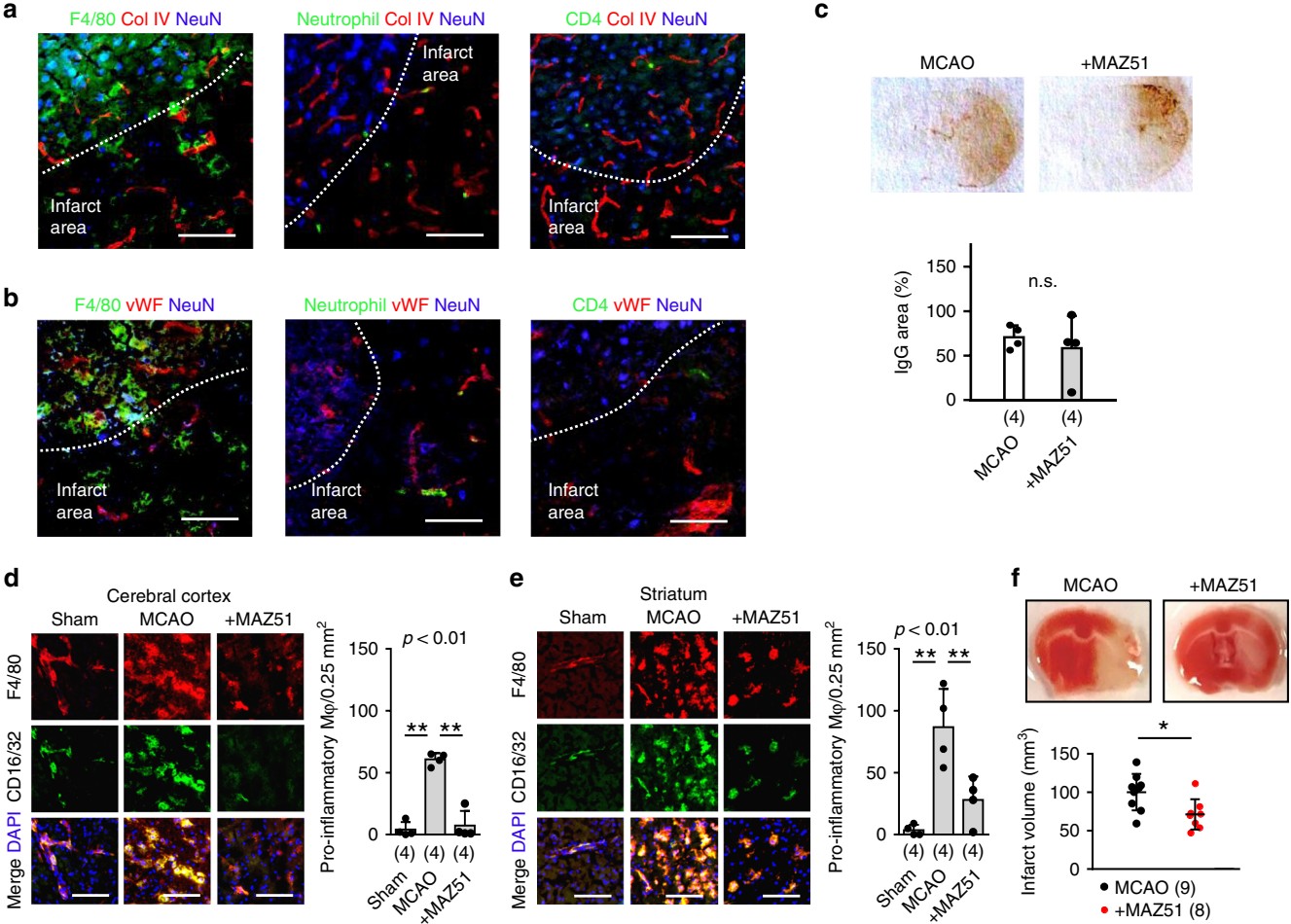

**Fig. 6** Effects of blocking cervical lymphatic activation in brain infarct formation after stroke: **a**, **b** Male C57BL6 mice were subjected to transient focal ischemia. Neuronal staining with NeuN antibody and basal lamina staining with collagen IV (Col IV) antibody or endothelial staining with vWF antibody were conducted to provide visual landmarks for immune cell infiltration including monocytes/macrophages, neutrophils or T-cells. Immunohistochemistry confirmed that monocytes/macrophages and neutrophils were accumulated in the ischemic area and peri-infarct cortex. A few T-cells were infiltrated in the same area at 72 h after stroke. Scale: 100 nm. **c** The percentage of IgG leaked area after MCAO was assessed. MAZ51 treatment did not influence IgG leakage after focal ischemia ($n = 4$ biologically independent animals). unpaired $t$-test. **d**, **e** CD16/32 antibody was used to identify pro-inflammatory macrophage phenotype. At 72 h after focal ischemia, pro-inflammatory macrophages were accumulated in ipsilateral cortex (**d**) and striatum (**e**). Intranasal treatment with MAZ51 (3 ng/10 μL) significantly reduced pro-inflammatory macrophages ($n = 4$ biologically independent animals). Scale: 100 nm. $**P < 0.01$, one-way ANOVA followed by Fisher's LSD test. **f** MAZ51 (3 ng/10 μL) significantly decreased cerebral infarction volumes at 72 h (MCAO; $n = 8$, MCAO + MAZ51; $n = 9$ biologically independent animals). Vehicle: $100.1 +/- 23.8$ mm$^3$, MAZ51: $71.2 +/- 19.7$ mm$^3$. $*P < 0.05$, unpaired $t$-test. All values are mean $+/-$ SD.

PBS twice, sections were incubated with ABC reagent for 30 min, followed by being washed with PBS for 10 min, incubated with detection substrate for 10 min. Infarct area after Nissl staining or IgG leaked area was calculated as the percentage over ipsilateral hemisphere by Image J software.

**CLN or ILN extraction.** Under isoflurane anesthesia the ventral neck area of mice were dissected and left and right superficial CLNs (cervical lymph nodes) were removed right before MCAO. As for ILN (inguinal lymph node) extraction, left and right ILNs were removed from the groin right before MCAO.

**Intracerebroventricular and intra-striatum administration.** Evans Blue dye (2%) was injected into lateral ventricles and intra-striatum in normal male Sprague Dawley (SD) rat. The rats were positioned on a stereotaxic frame and a 23-g stainless steel guide cannula was implanted into the right lateral ventricle using the stereotaxic coordinates from the bregma: −0.9 mm caudal, 1.4 mm lateral and 3.8 mm from the skull for i.c.v. injection and −0.2 mm caudal, 3.0 mm lateral and 6.0 mm for intra-striatum injection. The same procedure was used to inject (i.c.v.) MAZ51 (50 ng/5 μL) or respective vehicle (PBS) immediately after the stroke onset in a randomized fashion.

In normal male C57BL6 mice Evans Blue dye was injected into lateral ventricles or striatum at the following stereotaxic coordinates: i.c.v −0.5 mm caudal, 0.8 mm lateral, 2.5 mm deep. Striatum 0 mm caudal, 2.0 mm lateral, 3.5 mm deep.

**Intranasal injection.** Immediately after 60 min MCAO and reperfusion, when the mice were still deeply anaesthetized they were held by the ears and PBS (10 μL) or MAZ51 (3 ng/10 μL) were gradually released into the nostrils (5 μL in each nostril) with the help of a micropipette. We adjusted the rate of release to allow the mice to inhale the solution without trying to form bubbles.

**CSF collection.** Mice were flexed downward of the head at ~45 degree. A small incision was made in the skin. A capillary tube was insert into the citerna magna through the dura mater, lateral to the arteria dorsalis spinalis. When collecting CSF, the capillary tube was directly punctured into the cisterna magna, and the sample, without visible blood contamination, was collected into a tube. The same amount (5 μL) of CSF per mouse was used for western blot analysis.

**Primary lymphatic endothelial cultures.** Primary CLN lymphatic endothelial cultures were prepared from CLNs of C57Bl/6J mice (12–14 weeks). Briefly, CLNs were dissected, then cell suspensions were prepared. Biotin-conjugated LYVE-1 antibody (10 μg, NB100-725B, NOVUS biologicals) was mixed with CELLection Dynabeads (11533D, Thermo Fisher Scientific) coated with recombinant strepta-vidin (100 μl, $4 \times 10^7$ beads) for 30 min. After removing free floating antibody, beads were mixed with cell suspensions for 20 min at 4 °C. After the process of collecting the beads-bound cells and unbinding the beads, cells were plated and cultured with EBM2 plus supplement growth factors for 5–7 days.

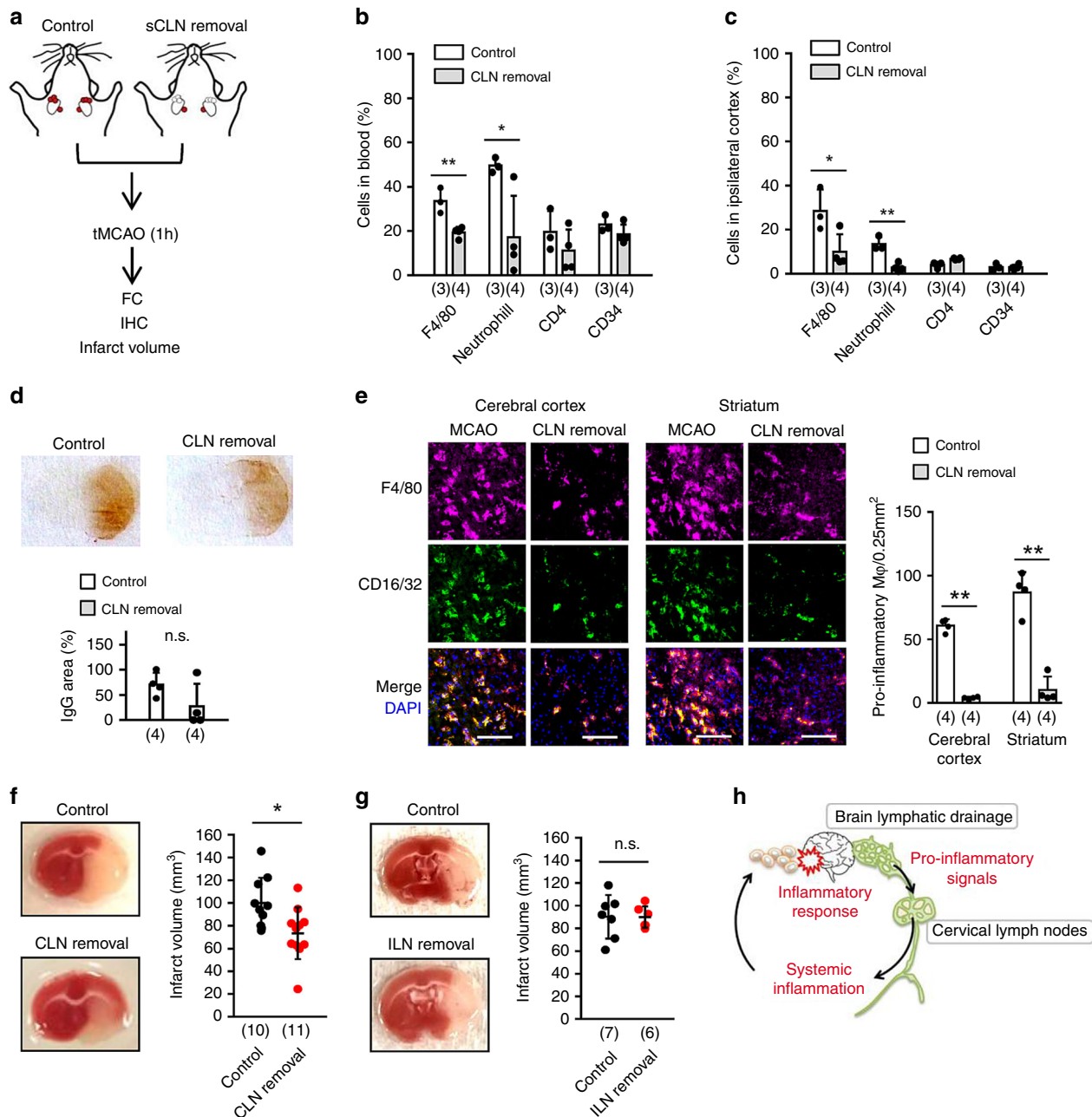

**Fig. 7** Superficial cervical node lymphadenectomy reduces brain damage after stroke: **a** Superficial CLNs were surgically removed from C57BL6 mice, then mice were subjected to transient 60 min focal cerebral ischemia. **b**, **c** Flow cytometry analysis demonstrated that accumulations of neutrophils and F4/80 positive monocytes/macrophages in both systemic circulation (**b**) and injured brain (**c**) were significantly reduced by superficial CLN removal (Control; $n =$ 3, CLN removal; $n = 4$ biologically independent animals). *$P < 0.05$, **$P < 0.01$, one-way ANOVA followed by Fisher's LSD test. **d**. Percentage of IgG leaked area in the ipsilateral hemisphere was assessed. CLN removal did not significantly influence IgG leakage after focal ischemia ($n = 4$ biologically independent animals). unpaired $t$-test. **e** CLN lymphadenectomy significantly decreased pro-inflammatory macrophages in ipsilateral cortex and striatum ($n = 4$ biologically independent animals), unpaired $t$-test. Scale: 100 nm. **f** CLN lymphadenectomy significantly reduced infarct volume at 72 h post-stroke (Control; $n = 10$, CLN removal; $n = 11$ biologically independent animals). Control: $99.9 +/- 22.3$ mm$^3$, CLN removal: $73.4 +/- 22.7$ mm$^3$. *$P < 0.05$, unpaired $t$-test. **g** Inguinal lymph nodes (ILN) lymphadenectomy did not decrease infarct volume at 72 h post-stroke (Control; $n = 7$, ILN removal; $n = 6$ biologically independent animals). Control: $90.2 +/- 19.1$ mm$^3$, ILN removal: $89.9 +/- 9.6$ mm$^3$, unpaired $t$-test. All values are mean $+/-$ SD. **h** Schematic of the proposed brain▪to-cervical lymph node pathway in the regulation of inflammatory response via VEGFR3 signaling after focal cerebral ischemia.

**Microarray analysis**. Mouse Gene 2.0ST CEL files were normalized to produce gene-level expression values using the implementation of the Robust Multiarray Average (RMA) in the *affy* package (version 1.36.1) included in the Bioconductor software suite (version 2.12) and an Entrez Gene-specific probeset mapping (17.0.0) from the Molecular and Behavioral Neuroscience Institute (Brainarray) at the University of Michigan. Array quality was assessed by computing Relative Log Expression (RLE) and Normalized Unscaled Standard Error (NUSE) using the *affyPLM* package (version 1.34.0). Human homologs of mouse genes were identified using HomoloGene (version 68). All microarray analyses were performed using the R environment for statistical computing (version 2.15.1).

**Gene Set Enrichment Analysis (GSEA)**. GSEA (version 2.2.1) was used to identify biological terms, pathways and processes that are coordinately up- or

downregulated within each pairwise comparison. The Entrez Gene identifiers of the human homologs of the genes interrogated by the array were ranked according to the moderated *t* statistic computed between 3H and sham groups. Mouse genes with multiple human homologs (or *vice versa*) were removed prior to ranking, so that the ranked list represents only those human genes that match exactly one mouse gene. This ranked list was then used to perform pre-ranked GSEA analyses (default parameters with random seed 1234) using the Entrez Gene versions of the Hallmark, Biocarta, KEGG, Reactome, Gene Ontology (GO), and transcription factor and microRNA motif gene sets obtained from the Molecular Signatures Database (MSigDB), version 5.0.

**Quantitative polymerase chain reaction analysis**. Total RNA were extracted using QIAzol lysis reagent (QIAGEN, 79306) from isolated lymphatic endothelial cells or peritoneal macrophages, followed by cDNA synthesis using High-capacity RNA-to-cDNA kit (ThermoFisher Seicntific, 4387406) according to the manufacturer's instructions. Relative levels of prox1 were determined by amplifying *PROX1* gene (Applied Biosystems, Mm00435969) and normalized by housekeeping gene *HPRT1* (Applied Biosystems, Mm01545399).

**Flow cytometry analysis**. Superficial CLNs or tissues collected from ipsilateral cortex including leptomeninges are gently minced and then digested at 37 °C for 30 min with an enzyme cocktail (Collagenase type IV; Sigma-Aldrich, C5138, DNase I; Sigma-Aldrich, D4263). FACS analysis was performed using a no labeled control for determining appropriate gates, voltages, and compensations required in multivariate flow cytometry.

**Western blot analysis**. Protein samples were prepared by Pro-PREPTM Protein Extraction Solution (INB17081, BOCA SCIENTIFIC). Each sample was loaded onto 4–20% Tris-glycine gels. After electorophresis and transferring to nitro-cellulose membranes, the membranes were blocked in Tris-buffered saline containing 0.1% Tween 20 and 0.2% I-block (T2015, Tropix) for 90 min at room temperature. Membranes were then incubated overnight at 4 °C with following primary antibodies. After incubation with peroxidase-conjugated secondary antibodies, visualization was enhanced by chemiluminescence (GE Healthcare, NA931-anti-mouse, or NA934- anti-rabbit, or NA935- anti-rat). Optical density was assessed using the NIH Image analysis software.

**Immunocytochemistry and immunohistochemistry**. Samples were initially fixed with 4% paraformaldehyde for 10 min at room temperature. Then, samples were processed with 0.1% Triton X for 5 min, followed by 3% BSA blocking for 1 h at room temperature. All primary antibodies were solved in 3% BSA. After staining with primary antibody overnight incubation at 4 °C, fluorescent-tagged secondary antibodies in 3% BSA were incubated for 1 h at room temperature, then nuclei were counterstained with 4,6-diamidino-2-phenylindole (DAPI), and coverslips were placed. Immunostaining images were obtained with a fluorescence microscope (Nikon ECLIPSE Ti-S) or Nikon A1SiR Confocal Microscope.

**Statistical analysis**. All graphs and statistical analysis were produced using GraphPad Prism 6. Results were expressed as mean ± SD. All of experiments were performed with full blinding, allocation concealment and randomization. When only two groups were compared, unpaired t-test was used. Multiple comparisons were evaluated by one-way ANOVA followed by Fisher's LSD test or repeated two-way ANOVA. $P < 0.05$ was considered to be statistically significant.

**Reporting summary**. Further information on research design is available in the Nature Research Reporting Summary linked to this article.

## Data availability

The datasets generated during and/or analyzed during the current study are available from the corresponding author on reasonable request. Raw data associated with Figs. 1b–d, 1f, h–i, 2a–f, 3a–c, 4b–c, j, l, 5f, i, 6c–f, 7b–g and Supplementary Figs. 1c, 2c, 6, 7d, 10b–c are available via a source data file submitted with this manuscript. Microarray data generated as part of Fig. 3e–g, and Supplementary Fig. 3 has been deposited in GEO Bank (accession number GSE138978).

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

## Acknowledgements

This work was supported in part by grants from NIH and the Rappaport Foundation. Cytometric assessments were supported by the Department of Pathology Flow and Image Cytometry Core. Microarray analysis was supported by the Boston University Microarray and Sequencing Resource Core Facility. X.J. was supported by the Chang Jiang Scholars Program (#T2014251) from the Chinese Ministry of Education.

## Author contributions

E.E., B.J.A., J.S., Y.N., J.H.P., E.T.M., Z.Y., S.J.C., R.D., and A.H. contributed to conducted experiments and data analysis. K.H. contributed to manuscript preparation, experimental design/analysis and conducted experiments. X.J. and E.H.L. contributed to manuscript preparation and experimental design.

## Competing interests

The authors declare no competing interests.
