## [Peer Review File · Nature Communications]

Reviewers' comments:

Reviewer #1 (Brain lymphatics, neuro-immune crosstalk)(Remarks to the Author):

Esposito et al. elucidate a proposed brain-to-cervical lymph node pathway that exacerbates a neuroinflammatory insult. They show that acute ischemic stroke activates immune cells in the cervical lymph node via Flt4 phosphorylation and lymphatic endothelial proliferation which cause the immune cells to enter the brain and drive further neuroinflammation. They also blocked this effect by either ICV injection of an Flt4 tyrosine kinase inhibitor or with surgical removal the cervical lymph nodes suggesting a causal role for the brain-cervical lymph node-brain signaling axis. They also demonstrate that the lymphatic endothelial cells of the cervical lymph node activate the immune cells after being stimulated by VEGF-C. While the entire mechanistic signaling story is incomplete, the authors present a clear, concise and novel finding that moves the field forward. We list moderate and minor issues that should be addressed below:

1. In Figure 1a, the authors only test for the presence of Evan's Blue in the cervical lymph node. They should also look for Evan's blue in other lymph nodes in the body to determine how specific this drainage is to cervical lymph nodes.
2. Regarding Figure 1E-G and 2A, we are uncertain about the physiological importance of proliferation of lymphatic endothelial cells in the context of driving peripheral inflammation. They only assess this via flow cytometry. Are the cells part of lymphatic vessels or individual cells interacting with immune cells? Are the lymphatic vessels sprouting new vessels or simply enlarging? The authors should stain for lymphatic vessels in the cervical lymph nodes to explain what is actually happening to these.
3. The authors state that there is an accumulation of macrophages in the lymph nodes. They should determine if this is migration into the lymph nodes or proliferation.
4. The authors need more markers for lymphatic endothelial cells in figure 1 and 2. They only use Flt4 in the rat experiments and LYVE-1 for the mouse experiments. Both markers potentially stain other cell populations and so the authors need to use multiple markers to demonstrate that lymphatic endothelial cells are increasing.
5. The B-actin blots in Figure 2A are way too overexposed. It is impossible to know whether the changes in LYVE-1 are actually do to changes in the protein expression vs. how much input protein was loaded.
6. There is a typo in the y axis of 3C. It should be "F4/80+ IL-1B +"
7. The conclusion in the results text of Figure 3F is overstated. Although the authors confirm that the brain-to-cervical lymph node signaling amplifies brain damage after cerebral ischemia in Figure 5, they can't confirm this from this experiment since the inhibitor was given systemically and thus could have effects on other lymph nodes, or cell types.
8. The control experiment in Figure 4F is incomplete. They only blot for iNOS but should also look at IL-1B and TGF-B as they did in 4D.
9. They need more readouts for stroke in Figure 5. The authors should stain for the different immune cells to determine where they are in the brain in respect to the infarct. They should also stain for neuronal death. In addition, blood-brain barrier leakage should also be assessed to determine how the immune cells are getting into the brain.

Reviewer #2 (Vascular biology, neuro-immune crosstalk)(Remarks to the Author):

The manuscript # NCOMMS-18-35445 entitled "Brain to cervical lymph node signaling after stroke" tests the potential hypothesis that brain-derived signals after ischemic stroke trigger proliferation of lymphatic endothelial cells and activation of the macrophages in the cervical lymph nodes to promote systemic inflammation. Moreover, the authors test the role that VEGF-C/VEGFR3 signaling plays in this process and they demonstrate that administration of MAZ inhibitor prevents proliferation of lymphatic endothelial cells and activation of the immune cells in cervical lymph nodes leading to smaller infarct volume. The findings of the manuscript are broadly quite interesting. However, there are a lot of gaps in the experimental design and inclusion of appropriate controls that reduce the scientific rigor of the study and question some of the findings.

Major concerns:

1. Throughout the paper, the authors talk about cervical lymph nodes. However, they do not distinguish between the superficial versus deep cervical lymph nodes. The CSF fluid drains in deep cervical nodes. In Figure 1, the authors examine proliferation of endothelial cells and immune cells in cervical lymph nodes. Which lymph nodes are these? Are these superficial or deep lymph nodes? The authors examine these changes at 24 hours after t-MCAO. I anticipate that there are a lot of systemic effects after 24 h of t-MCAO. Are these changes specific for the cervical lymph nodes? The authors need to include additional controls such as inguinal lymph nodes or gut lymph nodes to demonstrate that these changes are specific for cervical lymph nodes.
2. The data in Figure 1 would be more convincing if the authors show some immunofluorescence for the proliferation of endothelial cells. In addition, when does proliferation of lymphatic endothelial cells start in rats after t-MCAO? This needs to be addressed.
3. In Figure 2, the authors switch to mice and perform t-MCAO for 60 minutes. However, it is unclear why the timing of t-MCAO has now been switched from 100 min (rats) to 60 min (mice). The rationale was not clear. This is an important consideration since the timing of occlusion will affect the size of ischemic stroke and the timing when the damage occurs in the brain. The authors first need to show that the changes seen in Figure 1 in rats with 100 min t-MCAO are consistent in mice with a 60 minute t-MCAO. This is an important control to make sense and unify the data.
4. In Figure 2c and d, the authors need to validate some of the key transcriptome changes that they observe after 3 hours in cervical lymph nodes. Where are the excel data for transcriptome changes observed in Figures 2c and 2d? The authors may consider using annotated volcano plots to illustrate specific genes belonging to each pathway that change significantly in cervical lymph nodes after t-MCAO.
5. In Figure 3, is CCL2 the only chemokine that is changing in lymph nodes? How was this chemokine selected? The selection seems arbitrary. Do other chemokines known to induce migration of macrophages or other immune cells (e.g. T and B cells)?
6. The stroke volume reduction seems more robust with MAZ treatment than cervical lymph node removal (Fig. 5 and 6). These findings suggest that other lymphatic nodes may play a role in this process. The authors need to address this issue in their manuscript. Are other parameters beside stroke volume changing? Is the immune cell infiltration in the brain decreased? This is not clear in the

manuscript.

Minor Concerns:

1. Replace Flt-4 with the standard nomenclature name VEGFR3.

Reviewer #3 (Lymphatics, immune imaging)(Remarks to the Author):

In the manuscript by Esposito et al, the authors have investigated the effect of a stroke on the signaling in the downstream cervical lymph node. They have determined that within 24 h macrophages have accumulated and that tyrosine kinase pathways were active in LYVE1 positive sorted cells. Removal of the cervical lymph nodes or blockade of VEGFR3 tyrosine kinase signaling reduced the size of the brain infarct. The study is interesting as a first step in understanding how activation of the lymphatic system may lead to systemic response involving peripheral immune cells during stroke. However, the data is far too preliminary and leaves many important questions unanswered.

Major issues:

1. It is still completely unclear what signals are coming from the ischemia-injured brain that reach the lymph node. From what I can gather from the authors' data, the model is that these unknown signals activate first the lymph node LECs to induce a proliferation of these cells and then these cells activate the lymph node macrophages. However, one would expect that the signals would also directly affect the macrophages (CD169+) that are lining the subcapsular sinus of the lymph nodes. This is typically how antigens are processed in draining lymph nodes. The authors would need to more fully characterize the responses of both of the cell types at different time points after the induction of the model with a validated sorting strategy to phenotype these cell types.
2. The authors have used LYVE1 as the only marker for lymphatic endothelial cells throughout the manuscript. LYVE1 has been found to be expressed by activated macrophages in many models. The early upregulation of this marker (at 3 h) in Figure 2 is curious and unexpected as LYVE1 is typically considered to be downregulated during inflammatory challenge in LECS (see Johnson et al, JBC, 2007, doi: 10.1074/jbc.M702889200 and Vigil et al, Blood, 2011, doi: <https://doi.org/10.1182/blood-2010-12-326447>).
3. The authors need to provide evidence that the primary cervical lymph node lymphatic endothelial cells are truly LECs. As discussed above, LYVE-1 should not be used as the sole marker for sorting for LECs from the lymph nodes and one would expect after 5-7 days in culture that the cells have changed dramatically. Previous studies have used CD31/LYVE-1/podoplanin triple positivity and CD45/CD11b double negativity as the selection criterion for LECs in lymph nodes (Iftakhar-E-Khuda et al, PNAS, 2016 doi: 10.1073/pnas.1602357113). The expression of Prox1 should also be verified by PCR from the cultured cells to prove the LEC phenotype.
4. It is not apparent where and how the MAZ51 therapy is having an effect. If it is administered into the nasal cavity, is it blocking the expansion of lymphatic endothelial cells in the nasal mucosa? Is it reaching the subarachnoid space through the route of administration and possibly affecting the meningeal lymphatics? Or does it drain into the deep cervical lymph node and exert its effects there? This is not investigated or discussed in the manuscript.
5. A therapeutic approach to limit "lymphatic inflammation" at the early stages after stroke may not be a suitable strategy as activation of the systemic responses are necessary for the resolution of the injury. This is discussed briefly by the authors but it would be necessary to see how the infarction healing is affected by these interventions.

Minor issues:

The text is very limited and does not give a suitable overview of the previous literature regarding the lymphatic connections with the central nervous system in regards to antigen drainage (historical work of Helen Cserr and Roy Weller and more recent work from Jon Laman). There are also many details missing from the results and materials and methods that make it difficult to understand how the work was carried out.

The experiment involving injection of Evans blue into the lateral ventricles to demonstrate appearance of the dye in the draining cervical lymph node is not relevant to the current study. The "signals" from the stroke-affected brain would be coming from the parenchyma and the ventricle (CSF) to cervical lymph node pathways may just be one part of the pathway from the brain to the lymph nodes.

The authors mention in the discussion that "other lymphatic drainage routes such as the cribriform plate or mucosal lymphatics may also be involved". This is the major route for CNS drainage to the deep cervical lymph nodes, as has been verified in many previous studies.

The communication between the brain and the immune system mentioned in the same sentence to occur at the choroid plexus is not likely relevant to the activation of the peripheral immune system.

Reviewer 1:

1. In Figure 1a, the authors only test for the presence of Evan's Blue in the cervical lymph node. They should also look for Evan's blue in other lymph nodes in the body to determine how specific this drainage is to cervical lymph nodes.

Response: Thank you for this important suggestion. We have now assessed other lymph nodes including axillary and inguinal nodes after Evans blue ICV infusion. Figure 1 shows that the Evans blue tracer primarily drains into cervical nodes.

2. Regarding Figure 1E-G and 2A, we are uncertain about the physiological importance of proliferation of lymphatic endothelial cells in the context of driving peripheral inflammation. They only assess this via flow cytometry. Are the cells part of lymphatic vessels or individual cells interacting with immune cells? Are the lymphatic vessels sprouting new vessels or simply enlarging? The authors should stain for lymphatic vessels in the cervical lymph nodes to explain what is actually happening to these.

Response: Thank you for this suggestion. In addition to FACS, we now also perform immunostaining to confirm co-localization of LYVE-1 and Podoplanin or LYVE-1 and VEGFR3 in lymphatic endothelial cells

(as also suggested by Reviewer 3). After this, we perform double-staining using Podoplanin and Ki67 antibodies to assess lymphatic endothelial proliferation, which is a key aspect of lymphangiogenesis (e.g. Kukk et al, Development 1996; Fister et al, Blood 2010). Ki67 positive signals in the cervical nodes were found in lymphatic capillary-like structures at 3 hrs and in more individual cells at 24 hrs after focal cerebral ischemia. We also noticed that Podoplanin positive lymphatic endothelial cells were widely distributed within the cervical nodes post-stroke, as compared to the sham group. Taken together, these new data suggest that focal cerebral ischemia may increase cervical node lymphatic endothelial proliferation, which may be consistent with the well established phenomenon of lymphangiogenesis, as it has been widely described in the lymphatic inflammation literature. We now included the new data as well as key references in our revised manuscript.

3. The authors state that there is an accumulation of macrophages in the lymph nodes. They should determine if this is migration into the lymph nodes or proliferation.

Response: We apologize for being unclear. We now include immunostaining to show that CD169+ macrophages were not dramatically proliferated in the cervical nodes post-stroke. Instead, these cells appear to be activated and increase expression of IL-1beta after focal cerebral ischemia. Furthermore, flow cytometry analysis confirmed that the percentage of total macrophages was not significantly changed but IL-1beta positive macrophages were increased. Taken together, these data suggest that there is no substantial increase in macrophage numbers per se. However, it now appears that after stroke, VEGF-C to VEGFR signaling induces lymphangiogenesis in cervical nodes and secondarily activate these nodal macrophages. All these new data have been added.

4. The authors need more markers for lymphatic endothelial cells in figure 1 and 2. They only use Flt4 in the rat experiments and LYVE-1 for the mouse experiments. Both markers potentially stain other cell populations and so the authors need to use multiple markers to demonstrate that lymphatic endothelial cells are increasing.

Response: Thank you for this important suggestion. In our revised version, we now stain for all the additional markers, including CD31/LYVE-1/podoplanin triple positivity and CD45/CD11b double negativity (as also suggested by Reviewer 3).

5. The B-actin blots in Figure 2A are way too overexposed. It is impossible to know whether the changes in LYVE-1 are actually do to changes in the protein expression vs. how much input protein was loaded.

Response: We apologize for the suboptimal quality of our blots. We have reloaded and reblotted to improve the exposure. The new blots are now reanalyzed in the new Figure 3.

6. There is a typo in the y axis of 3C. It should be “F4/80+ IL-1B +”

Response: We apologize for the confusion. For FACS analysis in rats, we examined IL-1beta. But for FACS analysis in mice, we actually examined another pro-inflammatory cytokine marker, TNF-alpha, because this antibody works well in mice. Furthermore, by assessing these two different but prototypical inflammatory markers, we hoped to gain added confidence in the overall phenomenon of cervical node inflammation. However, to be sure, we now include protein quantification data for lymph nodes in the new Figure 4g-j, where we analyzed both IL-1beta and TNF-alpha in western blot. We hope that the reviewer will allow us to show data based on multiple inflammatory markers – this might actually increase overall confidence in the idea of “inflammatory activation”.

7. The conclusion in the results text of Figure 3F is overstated. Although the authors confirm that the brain-to-cervical lymph node signaling amplifies brain damage after cerebral ischemia in Figure 5, they can't confirm this from this experiment since the inhibitor was given systemically and thus could have effects on other lymph nodes, or cell types.

Response: We apologize for this overstatement in our original paper. We acknowledge that drug experiments may have systemic effects. Therefore, in this revision, we now include the surgical removal of non-cervical nodes as controls (as suggested by Reviewer 2 and the editor as well). With these new experiments, we confirmed that removal of cervical nodes reduced ischemic brain damage after stroke, whereas removal of inguinal lymph nodes did not ameliorate ischemic brain damage. Taken together, these findings may support the overall idea of brain-to-cervical node signaling after stroke.

8. The control experiment in Figure 4F is incomplete. They only blot for iNOS but should also look at IL-1B and TGF-B as they did in 4D.

Response: Thank you for these constructive comments. We now included blots for iNOS, IL-1beta, and TGF-beta.

9. They need more readouts for stroke in Figure 5. The authors should stain for the different immune cells to determine where they are in the brain in respect to the infarct. They should also stain for neuronal death. In addition, blood-brain barrier leakage should also be assessed to determine how the immune cells are getting into the brain.

Response: Thank you for raising these important points. In our revised manuscript, we have tried to assess all these additional endpoints, as suggested. First, we confirmed the distribution of ischemic injury with loss of NeuN staining, and then looked for IgG leakage to document the distribution of BBB damage. The spatial distributions of ischemic damage and BBB leakage were found to be consistent with what is expected for these MCAO models. Second, we performed co-labeling with multiple immune cell markers (F4/80, anti-neutrophil, CD4) to demonstrate that monocytes/macrophages and neutrophils were indeed present in ischemic and peri-infarct cortex, again as expected for these MCAO models. Importantly, immune cells appeared in close proximity to meningeal vessels, suggesting that meninges may be a vital source for immune cell infiltration. Further immunostaining in sagittal sections of the brain confirmed that immune cells were highly accumulated near meningeal vessels and choroid plexus. Our observation is consistent with emerging new data in the literature (Benakis et al, *Ther Adv Neurol Disord* 2018; Cai et al, *Nat Neurosci* 2019). We now included all the requested new data and references in our revised manuscript.

Reviewer 2:

1. Throughout the paper, the authors talk about cervical lymph nodes. However, they do not distinguish between the superficial versus deep cervical lymph nodes. The CSF fluid drains in deep cervical nodes. In Figure 1, the authors examine proliferation of endothelial cells and immune cells in cervical lymph nodes. Which lymph nodes are these? Are these superficial or deep lymph nodes? The authors examine these changes at 24 hours after t-MCAO. I anticipate that there are a lot of systemic effects after 24 h of t-MCAO. Are these changes specific for the cervical lymph nodes? The authors need to include additional controls such as inguinal lymph nodes or gut lymph nodes to demonstrate that these changes are specific for cervical lymph nodes.

Response: Thank you for raising these important points. In our revised manuscript, we assessed lymphatic endothelial proliferation in deep cervical lymph nodes (dCLNs), superficial cervical lymph nodes (sCLNs), and inguinal lymph nodes (ILNs) at 3 and 24 hrs after focal ischemia in rats. Our new data demonstrated that proliferating lymphatic endothelial cells were increased as early as 3 hrs in dCLNs and sCLNs. In contrast, lymphatic endothelium in ILNs were not significantly altered at 3 or 24 hrs after focal cerebral ischemia. All the new data are shown in new Figure 1. Altogether, these added controls (as well as the extra controls using surgical removal of inguinal nodes) should further support the idea of cervical node response after stroke.

2. The data in Figure 1 would be more convincing if the authors show some immunofluorescence for the

proliferation of endothelial cells. In addition, when does proliferation of lymphatic endothelial cells start in rats after t-MCAO? This needs to be addressed.

Response: Thank you for the suggestion (also suggested by Reviewer 1). As requested, we extracted lymph nodes at 3 and 24 hrs after focal cerebral ischemia in rats, and immunostaining with Podoplanin and Ki67 was performed to identify proliferating lymphatic endothelial cells. Our new data demonstrated that proliferating lymphatic endothelial cells were increased as early as 3 hrs in dCLNs and sCLNs. This acute response is now consistently observed in both rat as well as mouse models of focal cerebral ischemia.

3. In Figure 2, the authors switch to mice and perform t-MCAO for 60 minutes. However, it is unclear why the timing of t-MCAO has now been switched from 100 min (rats) to 60 min (mice). The rationale was not clear. This is an important consideration since the timing of occlusion will affect the size of ischemic stroke and the timing when the damage occurs in the brain. The authors first need to show that the changes seen in Figure 1 in rats with 100 min t-MCAO are consistent in mice with a 60 minute t-MCAO. This is an important control to make sense and unify the data.

Response: We apologize for not explaining our experimental rationale. Indeed, we intentionally selected tMCAO durations in order to achieve comparable ischemic injuries in mice and rats. In the very large experimental stroke literature, tMCAO is usually performed for durations between 60 to 120 min in rodents (e.g. Fluri et al, Drug Des Devel Ther 2015). However, responses tend to be generally more severe in mice compared to rats. In our laboratory, 60 min tMCAO in mice produces approximately 100 mm³ or 30% hemispheric infarction. For rats, approximately 30% hemispheric infarction is obtained after 100 min of tMCAO. Therefore, we selected these tMCAO durations to obtain equivalent stroke infarction in rats and mice, based on all our previous studies (e.g. Lan et al, Stroke 2018; Esposito et al, J Neurochem 2018; Esposito et al, Stroke 2015) as well as what has been previously reported by other research groups (e.g. Elvington et al, J Immunol 2012, Morris et al, PLoS One 2016).

4. In Figure 2c and d, the authors need to validate some of the key transcriptome changes that they observe after 3 hours in cervical lymph nodes. Where are the excel data for transcriptome changes observed in Figures 2c and 2d? The authors may consider using annotated volcano plots to illustrate specific genes belonging to each pathway that change significantly in cervical lymph nodes after t-MCAO.

Response: Thank you for this suggestion. As a part of our revision, we uploaded an excel file of our full dataset, including gene expression and GSEA analysis. Furthermore, as requested, we included annotated volcano plots in new Figure 3.

5. In Figure 3, is CCL2 the only chemokine that is changing in lymph nodes? How was this chemokine selected? The selection seems arbitrary. Do other chemokines known to induce migration of macrophages or other immune cells (e.g. T and B cells)?

Response: Thank you for raising this point. In our original paper, we selected CCL2 for analysis because it is a prototypical regulator of immune cell migration in stroke (e.g. Losy et al, Stroke 2001; Hughes et al, J Cereb Blood Flow Metab 2002), and our microarray analysis also showed that CCL2 was significantly upregulated in CLNs after stroke. However, we agree with the reviewer that this approach is somewhat “biased”. So, we went back and re-assessed our full dataset. It turns out that CCL28 emerged as the most dominantly upregulated chemokine in CLNs after focal cerebral ischemia. CCL28 is indeed known to regulate migration of lymphatic endothelial cells, myeloid cells and regulatory T cells (e.g. Kunkel et al, Immunity 2002; Mohan et al, Int Immunopharmacol 2017). We have now included references and the new CCL28 data in the new Figure 4. The original CCL2 data has been moved to the supplemental section.

6. The stroke volume reduction seems more robust with MAZ treatment than cervical lymph node removal (Fig. 5 and 6). These findings suggest that other lymphatic nodes may play a role in this process. The authors need to address this issue in their manuscript. Are other parameters beside stroke volume changing? Is the immune cell infiltration in the brain decreased? This is not clear in the manuscript.

Response: We apologize but hope it is ok to respectfully disagree. The degree of infarct reduction is equivalent in both mouse focal ischemia experiments. In the first experiment, infarct volume was $100.1 \pm 7.9 \text{ mm}^3$ (n=9) for vehicle versus $71.2 \pm 7.0 \text{ mm}^3$ (n=8) for MAZ51. In the second experiment, infarct volume was $99.9 \pm 7.0 \text{ mm}^3$ (n=10) for controls versus $73.4 \pm 6.9 \text{ mm}^3$ (n=11) for cervical node removal. These numbers are now explicitly stated in figure legends. Nevertheless, to be sure, we also examined immune cell infiltration in this revised paper. Our new data show that both MAZ51 and cervical node removal significantly decreased immune cell accumulation in ischemic brain hemisphere. Furthermore, we also showed that ILN (inguinal node) removal did not induce significant protection (extra control suggested by other reviewers and editor). All the new data are included in extended Figures 6 and 7.

Minor Concerns: Replace Flt-4 with the standard nomenclature name VEGFR3.

Response: We now use VEGFR3 in our revised manuscript.

Reviewer 3:

1. It is still completely unclear what signals are coming from the ischemia-injured brain that reach the lymph node. From what I can gather from the authors' data, the model is that these unknown signals activate first the lymph node LECs to induce a proliferation of these cells and then these cells activate the lymph node macrophages. However, one would expect that the signals would also directly affect the macrophages (CD169+) that are lining the subcapsular sinus of the lymph nodes. This is typically how antigens are processed in draining lymph nodes. The authors would need to more fully characterize the responses of both of the cell types at different time points after the induction of the model with a validated sorting strategy to phenotype these cell types.

Response: We are very grateful for this suggestion. We agree that our initial study did not fully document what might be the signal coming from ischemic brain to activate the cervical nodes. We know that lymphatic endothelial cells were potentially proliferated along with VEGFR3 tyrosine kinase phosphorylation. Because VEGF-C is a potent ligand for VEGFR3 and it is also known as a prototypical mediator to promote lymphangiogenesis (Kukk et al, Development 1996; Fister et al, Blood 2010), we looked at VEGF-C as our initial candidate signal. We collected cerebrospinal fluid (CSF) and found an increase of VEGF-C in rats and mice after focal ischemia, suggesting that brain-derived VEGF-C may be a potential factor coming from CSF to stimulate lymphatic endothelium in the cervical nodes. Additionally, as suggested by the reviewer, we now examined response in both lymphatic endothelial and macrophage cell populations. We performed immunostaining to assess CD169 positive macrophage activation at 3 and 24 hrs after focal cerebral ischemia. Interestingly, Ki67 positive signals were mainly found in Podoplanin positive lymphatic endothelial cells at 3 and 24 hrs after stroke. CD169 positive macrophages had a slight response at 3 hrs, but it was not statistically significant. However, IL-1beta and CD169 double-positive macrophages were clearly increased in CLNs at 24 hrs after stroke. Flow cytometry analysis revealed that the percentage of total macrophages was not significantly changed at 24 hrs. Collectively, after focal cerebral ischemia, lymphatic endothelial cells were rapidly proliferated early on (3 hrs) and then LN macrophages upregulated IL-1beta expression without substantial proliferation later on (24 hrs).

2. The authors have used LYVE1 as the only marker for lymphatic endothelial cells throughout the manuscript. LYVE1 has been found to be expressed by activated macrophages in many models. The

early upregulation of this marker (at 3 h) in Figure 2 is curious and unexpected as LYVE1 is typically considered to be downregulated during inflammatory challenge in LECS (see Johnson et al, JBC, 2007, doi: 10.1074/jbc.M702889200 and Vigl et al, Blood, 2011, doi: <https://doi.org/10.1182/blood-2010-12-326447>).

Response: Thank you for raising this point. Yes, we agree that Johnson et al and Vigl et al clearly showed internalization and degradation of LYVE-1 protein or decreasing LYVE-1, PROX1, VEGFR3 genes under inflammatory conditions. However, we believe that these inflammatory response will likely be complex, multifactorial and context-dependent. There may be three potential reasons why LYVE-1 protein or gene was upregulated in our models. First, lymphatic endothelial cells were proliferating in our models. Therefore, the increase of cell number may result in higher levels of LYVE-1 protein on western blots. Second (and most importantly), lymphatic endothelial response may be complex and dependent on context including timing, injury sites (local or distant), and brain damage severity. There are examples in the literature where other inflammatory conditions also increased LYVE-1 intensity and lymphatic endothelial proliferation (e.g. Yan et al, Sci Rep 2017; Wong et al, Nat Commun 2016). Third, as the reviewer correctly pointed out, LYVE-1 antibody may detect other cells besides lymphatic endothelial cells. Therefore, in order to confirm our findings in LYVE-1 positive cells, we performed immunostaining using another lymphatic endothelial marker, Podoplanin (as also suggested by other reviewers). Our initial assessment revealed that LYVE-1 positive area was well co-localized with Podoplanin positive area in CLNs regardless of whether stroke was present or not. Additionally, CD31 staining clearly separated blood vascular endothelial cells from lymphatic vessels stained with LYVE-1/Podoplanin, suggesting that our LYVE-1 antibody may be specific to lymphatic vessels per se. Furthermore, in vitro cell culture experiments showed that using same LYVE-1 antibody successfully isolated lymphatic endothelial cells (VEGFR3+, Podoplanin+, LYVE-1+, CD45-, CD11b-, Prox1 gene+) from CLNs (cervical nodes). We have included all these important points. Importantly, we agree with the reviewer that mechanisms are complex and dependent on disease conditions etc, so we also rigorously discuss all the potential caveats in our revised manuscript.

3. The authors need to provide evidence that the primary cervical lymph node lymphatic endothelial cells are truly LECs. As discussed above, LYVE-1 should not be used as the sole marker for sorting for LECs from the lymph nodes and one would expect after 5-7 days in culture that the cells have changed dramatically. Previous studies have used CD31/LYVE-1/podoplanin triple positivity and CD45/CD11b double negativity as the selection criterion for LECs in lymph nodes (Iftakhar-E-Khuda et al, PNAS, 2016 doi: 10.1073/pnas.1602357113). The expression of Prox1 should also be verified by PCR from the cultured cells to prove the LEC phenotype.

Response: Thank you for raising this point. As requested, we performed new experiments using CD31, LYVE-1, Podoplanin, CD11b, and CD45 to characterize lymphatic endothelial cells, and we confirmed that LYVE-1 positive cells indeed expressed VEGFR3, CD31, LYVE-1, Podoplanin, but they were negative for CD11b and CD45. We also tested Prox1 expression in qPCR analysis and confirmed that cells at 6 DIV following the isolation highly expressed Prox1 gene, but peritoneal macrophages barely expressed Prox1, suggesting that the isolated cells had lymphatic endothelium phenotype. These data are now included in the supplemental section.

4. It is not apparent where and how the MAZ51 therapy is having an effect. If it is administered into the nasal cavity, is it blocking the expansion of lymphatic endothelial cells in the nasal mucosa? Is it reaching the subarachnoid space through the route of administration and possibly affecting the meningeal lymphatics? Or does it drain into the deep cervical lymph node and exert its effects there? This is not investigated or discussed in the manuscript.

Response: Thank you for the suggestion. The inferior nasal turbinate contains numerous lymphatics and is connected to the nasolacrimal duct (NLD), which may further drain into superficial and deep cervical lymph nodes (Lohrberg and Wilting, *Cell Tissue Res* 2016). Indeed, it has been shown that intranasal antigen injection can induce immunological tolerance mediated by partly superficial CLNs (Wolvers et al, *J Immunol* 1999). In our initial experiments, we injected Evans blue into the nasal cavity and confirmed that the dye was rapidly accumulated in deep cervical lymph nodes and superficial cervical lymph nodes, suggesting that intranasal MAZ51 treatment can target both CLNs. Data are now included in supplemental section. This fundamentally (but not unequivocally) supports the idea of cervical node mechanisms. However, taken together with the other lines of evidence (e.g. surgical removal of cervical versus inguinal nodes), we still believe that brain-to-cervical node signaling should be an important (albeit not singular) pathway that can explain the “mystery” of how damaged brain sends out signals to the systemic response after stroke. We fully acknowledge that multiple sites of action may also be involved, and the caution suggested by this reviewer is correct, and we agree.

5. A therapeutic approach to limit "lymphatic inflammation" at the early stages after stroke may not be a suitable strategy as activation of the systemic responses are necessary for the resolution of the injury. This is discussed briefly by the authors but it would be necessary to see how the infarction healing is affected by these interventions.

Response: Thank you for reminding us of this important point. Indeed, our lab has always agreed with the biphasic nature of inflammation (e.g. Xing et al, *J Neuroinflamm* 2018; Choi et al, *Nat Med* 2016; Murata et al, *J Neurosci* 2012; Hayakawa et al, *PNAS* 2012; Lo, *Nat Med* 2010; Hayakawa et al, *Ann NY*

Acad Sci 2010; Lo, Nat Med 2008; Lee et al, J Neurosci 2006; Zhao et al, Nat Med 2006). So yes, our data may support a role for brain-to-cervical node signaling after stroke. But we agree that attempts to target this mechanism will have to be careful and nuanced. We now carefully include this important caveat in our paper. Nevertheless, we follow the reviewers advice and performed a whole new series of in vivo experiments to assess a wider range of endpoints for a longer period of time after focal cerebral ischemia in mice. These include infarct area, mortality/survival rate, and behavioral recovery following MAZ51 or cervical node removal during the recovery phase after stroke. Our additional experiments during the recovery phase revealed that both MAZ51 and cervical node removal decreased infarction and improved survival rate at day 8 post-stroke. There was no detectable change in behavioral testing. Mice are known to recover spontaneously after experimental stroke, and we see the same phenomenon here (Supplement figure 7). Although there were no statistically significant differences in behavioral recovery, it is important to note that both lymphatic interventions (MAZ51 or cervical node removal) did not worsen neurological outcome over time. This may suggest that acute lymphatic “treatments” may be still beneficial for preventing acute brain damage without interfering with beneficial aspects of delayed inflammation and remodeling. After stroke, lymphatic inflammation can be both deleterious (by augmenting the initial signal of systemic inflammation) or beneficial (since CNS lymphatic drainage system may support clearance of waste or debris produced by dying cells). Further studies are warranted to investigate the balance between these biphasic actions of lymphatic inflammation in stroke. We now include all our new in vivo experiments. We explicitly include all these fundamental points and caveats in our discussion. We fully agree with the reviewer that all aspects of inflammation will likely be biphasic with both good and bad. But we hope to have permission to report our current findings because we still feel that they may provide a novel answer to the question of how stroke-damaged CNS sends the initial signal out to the systemic response.

Minor issues: The text is very limited and does not give a suitable overview of the previous literature regarding the lymphatic connections with the central nervous system in regards to antigen drainage (historical work of Helen Cserr and Roy Weller and more recent work from Jon Laman). There are also many details missing from the results and materials and methods that make it difficult to understand how the work was carried out.

Response: Thank you for pointing out previous works related to our current study. We have tried our best to include the details in results (i.e. infarct size, ILN experiments) and materials and methods (catalog#, antibody dilution etc), We have included the suggested references in our revised manuscript.

Minor issues: The experiment involving injection of Evans blue into the lateral ventricles to demonstrate appearance of the dye in the draining cervical lymph node is not relevant to the current study. The

"signals" from the stroke-affected brain would be coming from the parenchyma and the ventricle (CSF) to cervical lymph node pathways may just be one part of the pathway from the brain to the lymph nodes.

Response: Thank you for the suggestion. In our revised manuscript, we additionally tested intra-striatum injection of Evans blue and confirmed that the tracer flowed into both deep and superficial CLNs. We have included the new data in the supplemental section. Finally, yes, we agree with the reviewer that brain parenchymal cells will surely produce many signaling factors after stroke. As suggested by the editor as well, we now have documented that VEGF-C is indeed present in the CSF after stroke and this may be a leading candidate to activate VEGFR3 in the cervical nodes. But we acknowledge that multiple factors may also be involved. Nevertheless, we still believe that the brain-to-cervical node pathway involving VEGF-C and VEGFR3 may be an important mechanism and respectfully seek permission to report this now.

Minor issues: The authors mention in the discussion that "other lymphatic drainage routes such as the cribriform plate or mucosal lymphatics may also be involved". This is the major route for CNS drainage to the deep cervical lymph nodes, as has been verified in many previous studies.

Response: We apologize for the confusion. We corrected these sentences.

Minor issues: The communication between the brain and the immune system mentioned in the same sentence to occur at the choroid plexus is not likely relevant to the activation of the peripheral immune system.

Response: We apologize and have rewritten this sentence.

Reviewers' comments:

Reviewer #1 (Remarks to the Author):

The authors largely answered the critiques and the paper is much more complete. There are still two minor points based on the previous questions:

Original reviewer 1, point 2 (very minor): It is great that the authors added the Ki67 experiment, and it is clear that the Ki67 colocalizes with the lymphatics in the stroke but not with the inhibitor. There is, however, Ki67 in other cells that the authors should comment on.

Original Reviewer 1, Point 9. The comment was meant for the authors to examine BBB dysfunction, immune cell infiltration and neuronal cell death comparing controls to perturbations (inhibitor or cervical lymph node removal). The authors only did these analyses for controls to show the strokes worked, however it would be good for them to do these analyses comparing the different cohorts.

Reviewer #2 (Remarks to the Author):

The revised manuscript # NCOMMS-18-35445A by Esposito et al., is significantly improved compared to the original submission. The authors have addressed the major concerns that I had with the original manuscript and have clarified some of the major questions that were unclear from the original manuscript. However, there are still some issues with the revised manuscript that the authors need to address to improve the clarity and strengthen the manuscript:

Major:

- 1) One of the major overall improvements of the manuscript is the addition of the images from the sCLNs and dSLNs. However, the resolution of these images is quite low throughout the paper (Figures 1, 2, 4 and Supplementary Figures 1 and 2). The authors are strongly urged to increase the magnification of these images and use high resolution confocal microscopy. The current images are NOT very convincing.
- 2) The Introduction needs to be expanded. It is missing a lot of important background information.
- 3) The Discussion needs to address the importance of VEGF-C/VEGFR3 signaling for lymphatic development in EAE that was recently published by Z. Fabry in Nature Communications. The larger implications of this study are not discussed in detail.
- 4) In Figure 6, the authors use Collagen IV as a marker for CNS blood vessels. This is incorrect. This is a basement membrane marker that changes in stroke. The authors need to use EC markers such as CD31, Glut1, Caveolin-1. In addition, they need to stain brain with Laminin to delineate the basal lamina. This will distinguish the immune cells that are perivascular versus those that have infiltrated the CNS.
- 5) In Figure legend 6, the author state that immune cells are in close proximity to meningeal vessels. How did they reach this conclusion? The images that they show do not support this since macrophages are scattered throughout the brain in the ipsilateral stroke region. This needs to be quantified and carefully examined.

Minor:

- 1) The manuscript contains several grammatical errors. The authors are urged to proofread the manuscript carefully.
- 2) The Supplementary Figures do not have titles. Please add titles to these figures.
- 3) In Page 4, the authors refer to lymphatic endothelial cell proliferation but cite the wrong figure. They cite Fig. 2b,c instead of Fig. 1b,c.
- 4) Where is the significance star for Figure 3a? That is missing.
- 5) The main and supplemental figure legends need to include the specific statistical test that was used to determine statistically-significant differences.

Reviewer #3 (Remarks to the Author):

The authors have addressed some of my initial concerns, however, the additional data has raised significant issues, particularly with the immunohistochemistry results. There are also many key experimental details that are still not included that are making it difficult for this reviewer to fully assess the data.

Major comments:

1. It is very difficult to appreciate that the stainings in the lymph nodes are really showing lymphatic vessels. For one, the images are tiny and are of not great quality in the PDF. It is also not clear where in the lymph nodes the images are taken from (e.g. subcapsular sinus, medullary region). Podoplanin is an unfortunate choice for the colocalization for Ki67, as podoplanin is highly expressed in the fibroblastic reticular cells of the lymph nodes. For example, in Figure 3b and c, the LYVE1 structures appear vessel-like but the podoplanin stained structures do not look like vessels at all.
2. Similarly, the images shown in Figure 4 for CD169 macrophages look nothing like the proper staining pattern. These macrophages are enriched at the subcapsular sinus, they should not be distributed throughout the lymph nodes. Please see many examples in the literature, e.g. Phan et al, Nat Immunol, 2009.
3. Many key experimental details are still missing in the methods. How were the injections of Evans Blue performed? What stereotactic coordinates were used in the mouse? No details are also given for intranasal administration. How was CSF harvested and assessed for levels of VEGFC? No details are also given for the IgG leakage study. The authors should go through their data figure by figure to assure that the experimental details are given for each type of experiment.

Minor comments:

1. "All values are mean +/- SEM." Values should be reported in mean +/- SD. Scatter plots would also make it easier to visualize the data.
2. Introduction: "It has been proposed that the brain may possess a lymphatic system" - the authors need to revise this statement. The brain itself has not been proposed to have a genuine lymphatic system. Furthermore, as the lymphatics that have been found in the dura mater are technically outside of the barriers of the CNS, there is some debate whether these vessels actually drain the CNS. What is clear is that the CNS does have lymphatic drainage, which has been shown in many studies to

occur through the cribriform plate and nasal mucosa. The authors should revise the statement to read something along the lines of "The CNS has a lymphatic drainage..."

3. Introduction: Lymphatic structures in meninges was described in adult mouse brains and this pathway may carry extracellular solutes into cervical nodes 12-14 - Reference 12 refers to the lymphatic system of paravascular structures - not lymphatics in the meninges.

4. Quantification appears to be missing in Figure 1A. In the figure legend it reads "Numbers represent accumulated Evans Blue amount (ng Evans Blue per LN)."

5. A volume of 10 uL is excessive for an injection into the striatum. With such a volume, it is certain that the bulk of the Evans Blue spills over into the CSF immediately. Please see the work of Helen Cserr and Paul Knopf, where maximum injection volumes were typically under 0.75 uL into one site of the parenchyma of rats to avoid "overflow to CSF and adjoining brain tissue" (Gordon et al, J. Neuroimmunology, 1992).

6. Why was VEGFR3 phosphorylation only checked for in superficial cervical lymph nodes, rather than the deep cervical lymph nodes?

7. Details are needed in Figure 2d figure legend. Which lymph nodes are assessed? Please add a label for the groups on the plots (white- sham / black - MCAO).

8. Figure 2f: "Immunohistochemistry confirmed that CD169 positive macrophages increased IL-1 β expression and MAZ51 treatment reduced IL-1 β positive macrophages." This is difficult to assess based on one representative image per condition, was any quantification performed to confirm the FACS data? How many mice are the images representing?

9. Results page 3: "Similar to rats, Evans blue dye tracing confirmed the presence of brain-to-CLN connection in C57BL6 mice." No figure is referred to here. What volume of EB was injected and what coordinates were used for the injections? If parenchymal injections were performed, then the same issue raised in point 5 would also be a concern.

10. Results, page 4: "VEGF-C appeared to be elevated in CSF after cerebral ischemia" Supp Fig 2 legend: "Western blot confirmed that VEGF-C was increased in CSF at 24 hours after focal cerebral ischemia in mice." Was any quantification made here? How many mice were assessed?

11. Results, page 4: "In deep and superficial CLNs, expression levels of the lymphatic endothelium marker LYVE-1 was increased after cerebral ischemia. (Fig. 3a)." The quantifications do not appear to show any significant changes in the deep cervical lymph nodes. What statistics were performed here? This information should be in the figure legends.

12. Results, page 4: "many other pharmacology studies have used this intranasal route to target cervical nodes 19,20" Reference 20 does not include any injection of a pharmacological compound.

13. Results, page 5: "consistent with an emerging literature supporting a role for meningeal vessels as a route for immune cell infiltration and post-stroke inflammation 24,25." It is unclear how Ref 25 is relevant to this topic - the paper discusses connections between the skull bone marrow and vessels in the dura mater, not the leptomeningeal vessels shown in the figure. The work of Alexander Flügel would be more relevant to the data.

14. Results, page 5: "Cervical lymph nodes were dissected" - which cervical LN groups are removed, superficial or deep or both? Again, some experimental details on the methods would be helpful here. The figure legend states that only superficial cervical lymph nodes were removed. There is quite some variability in CNS lymphatic outflow patterning between mice (see Ma et al, Nat Comm, 2017), but it is clear that deep cervical lymph nodes play a key role. How do the authors know that they removed all the relevant lymph nodes for CNS drainage?

15. Discussion: "ask whether other lymph nodes such as submandibular and parotid nodes may also be involved" - submandibular lymph nodes are the same as the superficial cervical lymph nodes (at least in mice). The nomenclature for lymph nodes varies depending on the authors.

16. Discussion: "Lymphatic endothelial cells were proliferated in our model, so the increase of cells may result in higher LYVE-1 levels." - I'm sorry, but this statement is unclear Wasn't the gene expression analysis was performed on sorted LECs? Isn't this adjusted to the number of cells?

17. Discussion: "Others have also demonstrated that lymphatic endothelial cells were increased during

lymphangiogenesis" - this is an obvious statement

18. Discussion: "Lymphatic drainage is able to refresh approximately 50% of CSF in rats, rabbits and sheep, but in humans, a significant portion of CSF may drain directly into the blood through arachnoid villi and granulations 41." This is very much a controversial topic. In mice, CSF appears to drain exclusively via lymphatics (Ma et al, Nat Comm, 2017). In humans, the question has never been answered directly.

Specific responses to each reviewer are listed below.

Reviewer #1 (Remarks to the Author):

Original reviewer 1, point 2 (very minor): It is great that the authors added the Ki67 experiment, and it is clear that the Ki67 colocalizes with the lymphatics in the stroke but not with the inhibitor. There is, however, Ki67 in other cells that the authors should comment on.

Response: We thank the reviewer for this important suggestion. We added this comment and caveat on other Ki67 positive cells, potentially including macrophages and T cells, in the figure legend.

Original Reviewer 1, Point 9. The comment was meant for the authors to examine BBB dysfunction, immune cell infiltration and neuronal cell death comparing controls to perturbations (inhibitor or cervical lymph node removal). The authors only did these analyses for controls to show the strokes worked, however it would be good for them to do these analyses comparing the different cohorts.

Response: We apologize for our misunderstanding. We have assessed immune cell infiltration (F4/80 and CD16) and neuronal damage (TTC staining and Nissl staining). As requested, we also analyzed IgG leakage areas in ischemic brain after MAZ51 treatment or cervical lymph node removal. We have included all these data in our revised manuscript.

Reviewer #2 (Remarks to the Author):

1) One of the major overall improvements of the manuscript is the addition of the images from the sCLNs and dSLNs. However, the resolution of these images is quite low throughout the paper (Figures 1, 2, 4 and Supplementary Figures 1 and 2). The authors are strongly urged to increase the magnification of these

images and use high resolution confocal microscopy. The current images are NOT very convincing.

Response: We thank the reviewer for this important suggestion. We have now included high resolution confocal microscopy images in Figures 1, 2, 4, S1 and S2.

2) The Introduction needs to be expanded. It is missing a lot of important background information.

Response: Thank you for this suggestion. We have expanded the Introduction.

3) The Discussion needs to address the importance of VEGF-C/VEGFR3 signaling for lymphatic development in EAE that was recently published by Z. Fabry in Nature Communications. The larger implications of this study are not discussed in detail.

Response: Thank you for pointing out this important new reference. We have cited and discussed this paper in our revised Discussion section.

4) In Figure 6, the authors use Collagen IV as a marker for CNS blood vessels. This is incorrect. This is a basement membrane marker that changes in stroke. The authors need to use EC markers such as CD31, Glut1, Caveolin-1. In addition, they need to stain brain with Laminin to delineate the basal lamina. This will distinguish the immune cells that are perivascular versus those that have infiltrated the CNS.

Response: Thank you for raising this point. Yes, we agree that Collagen IV is not a brain endothelial marker. We used collagen IV to help delineate vascular landmarks and provide visual context for showing the accumulation of inflammatory cells. We have now added endothelial vWF staining, as requested. However, we respectfully ask for permission not to completely re-do basal lamina staining. We agree with this reviewer that after stroke, vascular proteases such as MMPs and plasminogens begin to degrade the extracellular matrix. In fact, these basal lamina targets include collagen as well as laminin, perlecan etc (e.g. Haman et al, Stroke 1995; Fukuda et al, Stroke 2004). Therefore, adding laminin, perlecan etc stains may not add much, since all basal lamina substrates are being degraded and remodeled. Most importantly, we emphasize that our use of collagen (and now vWF) is only to provide landmarks for visual context. It is not our intent to dissect the complex phenomenon of how far infiltrating immune cells penetrate into the neurovascular unit. Indeed, this is almost an entire (and controversial) field unto itself, with complicated and diverse findings depending on model systems (e.g. Otxoa-de-Amezaga et al, Stroke 2019 vs Enzmann et al, Acta Neuropathol 2013). We have re-written our Results to clearly state that we only use immunostaining to provide visual context for immune cells in the brain. We moved Fig 6b-c into supplementary data. And we now explicitly acknowledge the ongoing controversy regarding immune cell penetration into different levels of the neurovascular unit, and we emphasize that this is not a focus of our study.

5) In Figure legend 6, the author state that immune cells are in close proximity to meningeal vessels. How did they reach this conclusion? The images that they show do not support this since macrophages are scattered throughout the brain in the ipsilateral stroke region. This needs to be quantified and carefully examined.

Response: Once again, we agree that these are complex questions, and whether and how far immune cells can migrate into brain after stroke is not the purpose of our study. Mechanisms of immune cell migration is a large and complicated field unto itself. We moved this tangential Figure into Supplementary Data. We apologize and sincerely hope that the reviewer may understand that this is not within the main scope of our study (please see response to point 4 above). Our main purpose is to show that CNS to cervical node signaling contributes to injury after stroke. Blocking VEGFR3 signaling or removal of cervical nodes (but not other nodes) reduced infarction after stroke. We respectfully suggest that, in the end, this is the most important data to support “causality” and demonstrate the fundamental importance of our brain-to-cervical-node signaling idea. We respectfully hope that the reviewer may allow us to focus on this main idea rather than the different question of how immune cells migrate.

Minor:

1) The manuscript contains several grammatical errors. The authors are urged to proofread the manuscript carefully.

Response: We apologize, and have tried our best to correct all grammatical errors.

2) The Supplementary Figures do not have titles. Please add titles to these figures.

Response: We have added titles to Supplementary Figures

3) In Page 4, the authors refer to lymphatic endothelial cell proliferation but cite the wrong figure. They cite Fig. 2b,c instead of Fig. 1b,c.

Response: We apologize and corrected this typo.

4) Where is the significance star for Figure 3a? That is missing.

Response: We are sorry for missing stars to show statistical significance. We now included all statistical results in each figure in our revised manuscript.

5) The main and supplemental figure legends need to include the specific statistical test that was used to determine statistically-significant differences.

Response: We have now specified all statistical tests in main and supplemental figure legends.

Reviewer #3 (Remarks to the Author):

1. It is very difficult to appreciate that the stainings in the lymph nodes are really showing lymphatic vessels. For one, the images are tiny and are of not great quality in the PDF. It is also not clear where in the lymph nodes the images are taken from (e.g. subcapsular sinus, medullary region). Podoplanin is an unfortunate

choice for the colocalization for Ki67, as podoplanin is highly expressed in the fibroblastic reticular cells of lymph nodes. For example, in Figure 3b and c, the LYVE1 structures appear vessel-like but the podoplanin stained structures do not look like vessels at all.

Response: We thank reviewer for this comment. We used podoplanin because this was a required comment from Reviewer 3 in the first round of review. We replaced all images with higher magnification and better quality in subcapsular sinus where lymphatic endothelial cells are located. We hope that overall images are now improved. To define co-localization, we used high resolution confocal microscope (as requested by Reviewer 2). However, we believe that our primary finding is best supported by the lymph node removal experiments. Removal of cervical nodes but not other nodes, reduced infarction. We respectfully suggest that this is the most important data to support “causality” and demonstrate the fundamental importance of our brain-to-cervical-node signaling idea.

2. Similarly, the images shown in Figure 4 for CD169 macrophages look nothing like the proper staining pattern. These macrophages are enriched at the subcapsular sinus, they should not be distributed throughout the lymph nodes. Please see many examples in the literature, e.g. Phan et al, Nat Immunol, 2009.

Response: We thank reviewer for this comment. We now include confocal microscopy to assess co-localization of cytokines in macrophages. We feel that our images are similar to those in Pham et al (please see below). We have tried our best and respectfully hope that our images are now acceptable.

[Redacted]

Phan et al, Nat Immunol, 2009.

[Redacted]

Esposito et al, 2019

3. Many key experimental details are still missing in the methods. How were the injections of Evans Blue performed? What stereotactic coordinates were used in the mouse? No details are also given for intranasal administration. How was CSF harvested and assessed for levels of VEGFC? No details are also given for the IgG leakage study. The authors should go through their data figure by figure to assure that the experimental details are given for each type of experiment.

Response: We apologize for being unclear as to the detail in methods. We tried our best to include all methods and details in our revised manuscript.

Minor comments:

1. "All values are mean +/- SEM." Values should be reported in mean +/- SD. Scatter plots would also make it easier to visualize the data.

Response: We changed SEM to SD, and all figures now also include scatter plots.

2. Introduction: "It has been proposed that the brain may possess a lymphatic system" - the authors need to revise this statement. The brain itself has not been proposed to have a genuine lymphatic system. Furthermore, as the lymphatics that have been found in the dura mater are technically outside of the barriers of the CNS, there is some debate whether these vessels actually drain the CNS. What is clear is that the CNS does have lymphatic drainage, which has been shown in many studies to occur through the cribriform plate and nasal mucosa. The authors should revise the statement to read something along the lines of "The CNS has a lymphatic drainage...".

Response: Thank you for this suggestion. We revised the introduction and tried to be more clear and specific.

3. Introduction: Lymphatic structures in meninges was described in adult mouse brains and this pathway may carry extracellular solutes into cervical nodes 12-14 - Reference 12 refers to the glymphatic system of paravascular structures - not lymphatics in the meninges.

Response: As requested, we have excluded this reference.

4. Quantification appears to be missing in Figure 1A. In the figure legend it reads "Numbers represent accumulated Evans Blue amount (ng Evans Blue per LN)."

Response: We apologize for the typo. We have corrected it.

5. A volume of 10 uL is excessive for an injection into the striatum. With such a volume, it is certain that the bulk of the Evans Blue spills over into the CSF immediately. Please see the work of Helen Cserr and Paul Knopf, where maximum injection volumes were typically under 0.75 uL into one site of the parenchyma of rats to avoid "overflow to CSF and adjoining brain tissue" (Gordon et al, J. Neuroimmunology, 1992).

Response: Thank you for this suggestion. We repeated the experiment with intra-striatum injection of 0.75 uL. At 3 hours, we still detected Evans Blue accumulation in cervical lymph nodes but not in inguinal lymph nodes. We have included these data in the supplemental section.

6. Why was VEGFR3 phosphorylation only checked for in superficial cervical lymph nodes, rather than the deep cervical lymph nodes?

Response: Thank you for raising this point. Yes, we tested superficial CLNs because we saw abundant Evans blue accumulation here. We also noticed that superficial CLNs had robust responses to ischemic

stroke in rats and mice (e.g. Figure 1b, 1c, figure 3a). Recent publication from the Nedargard lab has shown the drainage of brain lactate into superficial CLNs but not into inguinal LNs (J Cereb Blood Flow Metab. 2017 Jun; 37(6): 2112–2124). However, we agree that drainage occurs to both superficial and deep nodes. We now explicitly acknowledge this caveat in all sections of Results, Discussion and Figure Legends.

7. Details are needed in Figure 2d figure legend. Which lymph nodes are assessed? Please add a label for the groups on the plots (white- sham / black - MCAO).

Response: We now specify superficial CLNs. We included the missing label in Figure 2d.

8. Figure 2f: "Immunohistochemistry confirmed that CD169 positive macrophages increased IL-1 β expression and MAZ51 treatment reduced IL-1 β positive macrophages." This is difficult to assess based on one representative image per condition, was any quantification performed to confirm the FACS data? How many mice are the images representing?

Response: We are sorry for using low quality images. We now replaced representative images with high magnification/resolution and quantified the percentage of co-localization in n=4 independent experiment. We also revised our statements in the result section.

9. Results page 3: "Similar to rats, Evans blue dye tracing confirmed the presence of brain-to-CLN connection in C57BL6 mice." No figure is referred to here. What volume of EB was injected and what coordinates were used for the injections? If parenchymal injections were performed, then the same issue raised in point 5 would also be a concern.

Response: We are sorry for leaving out technical details. Please see revised Supplement Fig S2-a-b. For icv injection, Evans Blue dye (2%, 5 μ L) was injected into lateral ventricles (Anterior from bregma; -0.5 mm, Lateral from bregma; 0.8 mm, Depth; 2.5 mm). For intra-striatum injection, Evans Blue dye (2%, 0.2 μ L) was injected into striatum (Anterior from bregma; 0 mm, Lateral from bregma; 2.0 mm, Depth; 3.5 mm). All technical details are now included.

10. Results, page 4: "VEGF-C appeared to be elevated in CSF after cerebral ischemia" Supp Fig 2 legend: "Western blot confirmed that VEGF-C was increased in CSF at 24 hours after focal cerebral ischemia in mice." Was any quantification made here? How many mice were assessed?

Response: Thank you for the suggestion. We have added the quantification data.

11. Results, page 4: "In deep and superficial CLNs, expression levels of the lymphatic endothelium marker LYVE-1 was increased after cerebral ischemia. (Fig. 3a)." The quantifications do not appear to show any significant changes in the deep cervical lymph nodes. What statistics were performed here? This information should be in the figure legends.

Response: We are sorry for not including stars to show significant difference. We now included all statistical results in each figure and explained it in the figure legend.

12. Results, page 4: "many other pharmacology studies have used this intranasal route to target cervical nodes 19,20" Reference 20 does not include any injection of a pharmacological compound.

Response: We apologize for this incorrect citation. It has been replaced with the correct one.

13. Results, page 5: "consistent with an emerging literature supporting a role for meningeal vessels as a route for immune cell infiltration and post-stroke inflammation 24,25." It is unclear how Ref 25 is relevant to this topic - the paper discusses connections between the skull bone marrow and vessels in the dura mater, not the leptomeningeal vessels shown in the figure. The work of Alexander Flügel would be more relevant to the data.

Response: Thank you for the suggestion. We cited Ref 25 (Cai et al., Nat Neurosci. 2019 Feb;22(2):317-327) because authors discussed the possibility of meningeal contribution for immune cell entry and/or exit routes of the brain. As per this reviewer's recommendation, we now also include the Flügel paper in our revised manuscript.

14. Results, page 5: "Cervical lymph nodes were dissected" - which cervical LN groups are removed, superficial or deep or both? Again, some experimental details on the methods would be helpful here. The figure legend states that only superficial cervical lymph nodes were removed. There is quite some variability in CNS lymphatic outflow patterning between mice (see Ma et al, Nat Comm, 2017), but it is clear that deep cervical lymph nodes play a key role. How do the authors know that they removed all the relevant lymph nodes for CNS drainage?

Response: Thank you for raising this issue. As explained above, we removed the superficial nodes for an operational reason (we saw abundant Evans Blue dye accumulation in these nodes and we also detected significant activation response after stroke) as well as a literature-based reason (findings from the Nedegaard lab). In the end, we found that surgical removal of superficial nodes (but not inguinal nodes) significantly decreased infarct volumes. In the end, this result may still support the overall idea, i.e. brain-to-cervical node signaling may worsen injury. Another operational reason is related to practical problems with surgery for deep nodes. Deep cervical node extraction often caused bleeding, which must be avoided in stroke studies. What remains very important is that removing superficial cervical nodes can still significantly influence stroke outcome. We have added these important caveats of the need for future studies to more carefully assess deep node mechanisms. But we respectfully hope that the reviewer may still allow us to report our current findings.

15. Discussion: "ask whether other lymph nodes such as submandibular and parotid nodes may also be involved" - submandibular lymph nodes are the same as the superficial cervical lymph nodes (at least in mice). The nomenclature for lymph nodes varies depending on the authors.

Response: We apologize for this typo. It has been corrected.

16. Discussion: "Lymphatic endothelial cells were proliferated in our model, so the increase of cells may result in higher LYVE-1 levels." - I'm sorry, but this statement is unclear. Wasn't the gene expression analysis was performed on sorted LECs? Isn't this adjusted to the number of cells?

Response: We apologize for being unclear. This statement was meant to refer to immunohistochemistry. We have re-written the sentence for clarity.

17. Discussion: "Others have also demonstrated that lymphatic endothelial cells were increased during lymphangiogenesis" - this is an obvious statement

Response: Thank you. We deleted this sentence.

18. Discussion: "Lymphatic drainage is able to refresh approximately 50% of CSF in rats, rabbits and sheep, but in humans, a significant portion of CSF may drain directly into the blood through arachnoid villi and granulations 41." This is very much a controversial topic. In mice, CSF appears to drain exclusively via lymphatics (Ma et al, Nat Comm, 2017). In humans, the question has never been answered directly.

Response: Thank you for raising this point. We have re-written this sentence.

Reviewers' comments:

Reviewer #1 (Remarks to the Author):

The authors addressed my concerns

Reviewer #2 (Remarks to the Author):

The revised manuscript # NCOMMS-18-35445B has significantly improved compared to the previous submission. The authors have made a strong effort to improve the clarity of the lymph node analysis by providing confocal images to visualize distinct cell types throughout the manuscript. In addition, they have taken into consideration the majority of the concerns that I expressed with the previous submission.

There are a few minor issues need to be addressed to improve the manuscript:

- 1) The manuscript needs to be CAREFULLY read and edited by a native English speaker. It contains multiple grammatical errors that need to be corrected. The authors are strongly urged to give the manuscript to a native English speaker for proofreading.
- 2) The introduction needs to provide more details about the rationale for the study in stroke. Some of the information presented in discussion regarding the stroke needs to be presented in the introduction and elaborated more in the discussion.
- 3) The title of Figure S1 and S2 do not capture the main findings of the figure. These need to be revised.
- 4) Please add a figure legend with details for Figure S3.

Reviewer #3 (Remarks to the Author):

In the revised manuscript by Esposito et al, the authors have addressed some of the concerns that were raised during the previous round of submission. However, I still have major reservations regarding the presented immunostaining of lymphatic vessels and CD169 macrophages within the lymph node.

1. The authors have continued to use podoplanin for the staining of the lymphatic vessels in the cervical lymph nodes. They cite a comment from this reviewer as the reasoning for this. However, the comment was referring to FACS studies, where the use of anti-podoplanin antibodies is well-established in conjunction with anti-CD31 to sort or analyze LECs. In immunohistological sections, it is absolutely not possible to identify lymphatic vessels with only podoplanin staining due to its expression on several types of stromal cells in the node. Please see the recent resource paper from Jason Cyster on stromal cells in the lymph node (Figs. 1 and 2):

<https://www.sciencedirect.com/science/article/pii/S1074761318301432> .

LYVE-1 is the preferred antibody to immunostain lymphatic vessels in lymph nodes. The vessels can be easily confirmed by their morphology and can be quantified for lymphatic area. Single-cells, which

would likely represent macrophages, can be excluded from this analysis. Please see the many examples from the literature, e.g.

<https://www.jimmunol.org/content/188/8/4065>

2. The authors claim that "CD169-positive macrophages in subcapsular sinus were modestly proliferated (Fig. 2a-c)." and that "CD169-positive macrophages in subcapsular sinus also strongly expressed IL-1 β " in Fig. 2f. However, the quantified regions are not at the subcapsular sinus, but appear to be located deeper within of the lymph nodes. The images are still not of high enough resolution (at least in the PDF) to judge the quality of the stainings.

Minor comments:

Please relocate the comment that is within Figure 6 legend to either the results or discussion section. "NOTE: immune cells were also found in close proximity to leptomeninges, choroid plexus, and brain parenchyma in the ipsilateral hemisphere (Supplemental Fig. 9), consistent with an emerging literature supporting a role for meningeal vessels as a route for immune cell infiltration and post-stroke inflammation 58-60."

There are still several grammatical errors and the whole manuscript would benefit from a close proofreading.

Specific responses to each reviewer are listed below.

Reviewer #1 (Remarks to the Author):

The authors addressed my concerns

Response: We thank the reviewer for this supportive comment.

Reviewer #2 (Remarks to the Author):

1) The manuscript needs to be CAREFULLY read and edited by a native English speaker. It contains multiple grammatical errors that need to be corrected. The authors are strongly urged to give the manuscript to a native English speaker for proofreading.

Response: We are sorry for grammatical errors in our manuscript. We have now tried our best to carefully check and proofread everything with a native English speaker.

2) The introduction needs to provide more details about the rationale for the study in stroke. Some of the information presented in discussion regarding the stroke needs to be presented in the introduction and elaborated more in the discussion.

Response: Thank you for this suggestion. As requested, we have now moved sections related to the rationale of inflammation from the Discussion section to the Introduction. We have also expanded the Discussion.

3) The title of Figure S1 and S2 do not capture the main findings of the figure. These need to be revised.

Response: We apologize, and have revised figure legends.

4) Please add a figure legend with details for Figure S3.

Response: We added a figure legend for Figure S3.

Reviewer #3 (Remarks to the Author):

1. The authors have continued to use podoplanin for the staining of the lymphatic vessels in the cervical lymph nodes. They cite a comment from this reviewer as the reasoning for this. However, the comment was referring to FACS studies, where the use of anti-podoplanin antibodies is well-established in conjunction with anti-CD31 to sort or analyze LECs. In immunohistological sections, it is absolutely not possible to identify lymphatic vessels with only podoplanin staining due to its expression on several types of stromal cells in the node. Please see the recent resource paper from Jason Cyster on stromal cells in the lymph node (Figs. 1 and 2): <https://www.sciencedirect.com/science/article/pii/S1074761318301432> . LYVE-1 is the preferred antibody to

immunostain lymphatic vessels in lymph nodes. The vessels can be easily confirmed by their morphology and can be quantified for lymphatic area. Single-cells, which would likely represent macrophages, can be excluded from this analysis. Please see the many examples from the literature, e.g.

<https://www.jimmunol.org/content/188/8/4065>

Response: We thank reviewer for this comment. As requested, we replaced all Podoplanin images with LYVE-1 staining in rat LNs in Figure 1. We agree with reviewer that LYVE-1 staining has been well documented in mice. The literature in rats may not be as extensive. However, we found a few prominent papers showing LYVE-1 staining in LNs (Ohtani et al., Arch Histol Cytol. 2008;71:69-76; Hadamitzky et al., J Vasc Res. 2009;46:389-396). Our LYVE-1 images are very comparable to these published images.

[Redacted]

[Redacted]

Esposito et al., 2019

Ohtani et al., 2008

[Redacted]

Hadamitzky et al., 2009

Then, we analyzed LYVE-1 and Ki67 positive cells in the area of subcapsular sinus. We did not count single-cell LYVE-1 and Ki67 positive cells in order to exclude potential macrophage population. As requested by the reviewer, we have now included these data in new Figure 1.

2. The authors claim that "CD169-positive macrophages in subcapsular sinus were modestly proliferated (Fig. 2a-c)." and that "CD169-positive macrophages in subcapsular sinus also strongly expressed IL-1 β " in Fig. 2f. However, the quantified regions are not at the subcapsular sinus, but appear to be located deeper within of the lymph nodes. The images are still not of high enough resolution (at least in the PDF) to judge the quality of the stainings.

Response: We apologize and thank reviewer for this comment. We have redone all the relevant figures for the subcapsular sinus region (previously Figure 2a-c and Figure 2f) in order to improve their magnification. We sincerely hope that overall images are now improved.

Minor comments:

Please relocate the comment that is within Figure 6 legend to either the results or discussion section. "NOTE: immune cells were also found in close proximity to leptomeninges, choroid plexus, and brain parenchyma in the ipsilateral hemisphere (Supplemental Fig. 9), consistent with an emerging literature supporting a role for meningeal vessels as a route for immune cell infiltration and post-stroke inflammation 58-60."

Response: Thank you for this suggestion. We have moved this sentence to the Results section, as requested.

There are still several grammatical errors and the whole manuscript would benefit from a close proofreading.

Response: We apologize. Our manuscript has now been carefully checked and proofread by a native English speaker.

REVIEWERS' COMMENTS:

Reviewer #3 (Remarks to the Author):

In the revised manuscript by Esposito et al, the authors have addressed the major concerns that were raised during the previous round of submission.

1. The authors have provided LYVE-1 staining of the lymphatic vessels in the rat lymph node. This staining appears to now truly be representing lymphatic vessels, unlike the podoplanin staining that was provided in the last round of revision. One comment regarding the quantification of Ki67/LYVE-1 cells, the authors have mentioned in the comments to the reviewer that "We did not count single-cell LYVE-1 and Ki67 positive cells in order to exclude potential macrophage population." This makes little sense as these Ki67 positive cells by nature would be single cells. This reviewer's previous comment regarding exclusion of single-cells was referring to quantifications of LYVE1 positive lymphatic area. In future, the authors are recommended to use Prox1/Ki67 double staining for quantifications of proliferation of lymphatic endothelial cells.
2. The expanded introduction refers to many reports from the literature on hemorrhagic stroke rather than ischemic stroke. The authors are recommended to focus their introduction and discussion on systemic inflammation on ischemic stroke only. Also, the authors have mentioned "PPAR γ agonist" and "N2" phenotypes without any introduction to these concepts.
3. In the discussion Line 211: "removal of the cervical lymph nodes but not other nodes" the authors again need to make clear that only superficial lymph nodes were removed and not both deep and superficial lymph nodes.

Reviewer #3 (Remarks to the Author):

1. The authors have provided LYVE-1 staining of the lymphatic vessels in the rat lymph node. This staining appears to now truly be representing lymphatic vessels, unlike the podoplanin staining that was provided in the last round of revision. One comment regarding the quantification of Ki67/LYVE-1 cells, the authors have mentioned in the comments to the reviewer that "We did not count single-cell LYVE-1 and Ki67 positive cells in order to exclude potential macrophage population." This makes little sense as these Ki67 positive cells by nature would be single cells. This reviewer's previous comment regarding exclusion of single-cells was referring to quantifications of LYVE1 positive lymphatic area. In future, the authors are recommended to use Prox1/Ki67 double staining for quantifications of proliferation of lymphatic endothelial cells.

Response: We thank the reviewer for this supportive comment. We agree that use of multiple lymphatic markers including Prox1 will strengthen any findings related to lymphatic endothelial response. As suggested, we have now included this as a potential caveat to be considered in future study in the discussion section.

2. The expanded introduction refers to many reports from the literature on hemorrhagic stroke rather than ischemic stroke. The authors are recommended to focus their introduction and discussion on systemic inflammation on ischemic stroke only. Also, the authors have mentioned "PPAR γ agonist" and "N2" phenotypes without any introduction to these concepts.

Response: Thank you for this suggestion. As requested, we now cite literature for ischemic stroke in the revised Introduction and Discussion. We also modified the sentence regarding neutrophil activation.

3. In the discussion Line 211: "removal of the cervical lymph nodes but not other nodes" the authors again need to make clear that only superficial lymph nodes were removed and not both deep and superficial lymph nodes.

Response: We are sorry for being unclear. We have revised the sentence to explicitly state that superficial nodes were removed.